

# The Contribution of Fires to TES Observations of Free Tropospheric PAN over North America in July

Emily V. Fischer[1], Liye Zhu[1], Vivienne H. Payne[2], John R. Worden[2], Zhe Jiang[4], Susan S. Kulawik[3], Steven Brey[1], Arsineh Hecobian[1], Daniel Gombos[7], Karen Cady-Pereira[5], and Frank Flocke[6]

[1]Department of Atmospheric Science, Colorado State University, Fort Collins, CO, USA
[2]Jet Propulsion Laboratory, California Institute of Technology, Pasadena, CA, USA
[3]Bay Area Environmental Research Institute Moffett Field, Moffett Field, CA, USA
[4] National Center for Atmospheric Research, Boulder, CO, USA
[5] Atmospheric and Environmental Research (AER), Lexington, MA, USA
[6] National Center for Atmospheric Research (NCAR), Boulder, CO, USA
[7] MORSE Corp, Cambridge, MA, USA

*Correspondence to*: Emily V. Fischer (evf@atmos.colostate.edu)

**Abstract.** Peroxyacetyl nitrate (PAN) is a critical atmospheric reservoir for nitrogen oxide radicals, and it plays a lead role in their redistribution in the troposphere. We analyze new Tropospheric Emission Spectrometer (TES) PAN observations over North America during July 2006 to 2009. Using aircraft observations from the Colorado Front Range, we demonstrate that TES can be sensitive to elevated PAN in the boundary layer even in the presence of clouds. In situ observations have shown that wildfire emissions can rapidly produce PAN, and PAN decomposition is an important component of ozone production in smoke plumes. We identify smoke-impacted TES PAN retrievals by co-location with NOAA Hazard Mapping System (HMS) smoke plumes. We find that 15 – 32 % of cases where elevated PAN is identified in TES observations (retrievals with DOF > 0.6) overlap smoke plumes. A case study of smoke transport in July 2007 illustrates that PAN enhancements associated with HMS smoke plumes can be connected to fire complexes, providing evidence that TES is sufficiently sensitive to measure elevated PAN several days downwind of major fires. Using a subset of retrievals with TES 510 hPa carbon monoxide (CO) > 150 ppbv, and multiple estimates of background PAN, we calculate enhancement ratios for tropospheric average PAN relative to CO in smoke-impacted retrievals. Most of the TES-based enhancement ratios fall within the range calculated from in situ measurements.





## 1 Introduction

PAN is considered to be the largest reservoir for nitrogen oxide radicals ($NO_x = NO + NO_2$) in the
troposphere, and it plays a major role in the redistribution of $NO_x$ from sources to remote regions (Singh,
1987; Singh and Hanst, 1981). The balance between ozone ($O_3$) production and destruction is dictated by
the abundance of $NO_x$ (Monks et al., 2015), and thus the distribution of $O_3$ is a function of PAN production,
transport, and decomposition rates (Kasibhatla et al., 1993; Moxim et al., 1996; Wang et al., 1998).
However, due to the complexity of its formation chemistry and its sensitivity to vertical transport (Fischer
et al., 2014) PAN is difficult to represent in global chemical transport models (CTMs) (Emmons et al.,
2015), and in plume scale models (Alvarado et al., 2015).

In situ observations from aircraft show rapid conversion of $NO_x$ to PAN in smoke plumes
(Alvarado et al., 2010; Müller et al., 2016) seemingly due to the oxidation of relatively short-lived non-
methane volatile organic compounds (NMVOCs), particularly oxygenated species emitted in higher
quantities. Elevated PAN in smoke plumes can travel significant distances (Lindaas et al., 2017), the $NO_x$
that is eventually released can contribute to $O_3$ production (Bein et al., 2008; Brey and Fischer, 2016; Jaffe
et al., 2013; Lindaas et al., 2017; Morris et al., 2006; Pfister et al., 2008; Singh et al., 2012), but models are
unlikely to accurately predict fire-related $O_3$ without better incorporating the evolution of PAN in the
smoke (Jaffe et al., 2013). Efforts to understand the abundance and distribution of PAN related to smoke
over North America are timely because the area burned by wildfires in the western U.S. has increased in
recent decades (Westerling, 2016; Westerling et al., 2006), and though there is spread in the predictions,
fire activity is expected to continue to increase over the coming decades (Hurteau et al., 2014; Keywood et
al., 2013; Moritz et al., 2012; Scholze et al., 2006; Yue et al., 2013). In addition, anthropogenic $NO_x$
emissions are declining over most of North America (Pinder et al., 2008), implying that wildfires could
have a greater relative impact on U.S. air quality in the future (Val Martin et al., 2015).

Aside from a handful of long term observational datasets (e.g. Brice et al. (1988); Pandey Deolal
et al. (2014); Fischer et al. (2011); Tanimoto et al. (2002); Mills et al. (2007)), much of our understanding
of the distribution of PAN outside urban areas rests on data from aircraft missions interpreted with global
chemical transport models (Alvarado et al., 2010; Fischer et al., 2014). Given the limited set of long-term
in situ measurements, satellite measurements are essential to understand the seasonal cycle and interannual
variability of PAN in the troposphere along with which processes contribute to these features. Limb-
sounding satellite instruments have provided global distributions of PAN in the upper troposphere and
lower stratosphere (Glatthor et al., 2007; Moore and Remedios, 2010; Wiegele et al., 2012). Analysis of
new observations of PAN from the Tropospheric Emission Spectrometer (TES) can be used to look lower
in the troposphere (Payne et al., 2014). TES PAN observations confirm the important role that high latitude
fires play in the interannual variability of PAN during spring at high latitudes (Zhu et al., 2015), support
estimates of the role of PAN in the transpacific transport of $O_3$ (Jiang et al., 2016), establish strong
intercontinental transport of PAN in both spring and summer (Zhu et al., 2017), and provide confirmation





of PAN features in the tropics predicted by CTMs (Payne et al., 2016). TES retrievals have also shown
elevated PAN in smoke plumes over North America (Alvarado et al., 2011).

Here we present an analysis of TES PAN observations over North America during the month of
July between 2006 and 2009. We focus on understanding the contribution of smoke to enhanced PAN by
segregating TES PAN retrievals based on smoke-impact through comparisons to NOAA Hazard Mapping
System (HMS) smoke plumes.

**2 Methods**

**2.1 TES PAN observations**

TES is a nadir-viewing Fourier transform spectrometer that measures thermal infrared radiances at
a high spectral resolution ($0.1$ $cm^{-1}$ apodized), and it is one of four instruments on the NASA Aura satellite,
which flies in a sun-synchronous orbit with local equator crossing times of 1:30 and 13:30. TES has a
number of observational modes (global survey, and special observation modes such as step-and-stare and
transect). In global survey mode TES makes measurements along the satellite track for 16 orbits with a
spacing of ~200 km; in step-and-stare mode nadir measurements are made every 40 km along the track for
approximately 50 degrees of latitude; in transect mode observations consist of series of 40 consecutive
scans spaced 12 km apart.

Specific details of the TES PAN retrieval algorithm are provided in Payne et al. (2014). TES PAN
retrievals are being processed routinely for the whole TES dataset and are publicly available in the TES v7
Level 2 product. However, at the time of this work, the v7 product was not yet available. The TES PAN
retrievals shown here were processed using a prototype algorithm for the area and time periods of interest.
On a single footprint basis, TES is capable of measuring elevated PAN (detection limit ~ 0.2 ppbv) in the
free troposphere, with uncertainty of 30-50 %. The PAN spectral feature at 1140-1180 $cm^{-1}$ used for the
TES retrievals coincides with the location of a silicate feature in surface emissivity spectra. For footprints
where the spectra show strong evidence of this silicate feature in the surface emissivity (this can occur over
rocky or sandy surfaces), TES PAN retrievals are not attempted. In order to illustrate the characteristics of
the retrievals, the four panels in Figure 1 show simulated retrievals for different combinations of
conditions. The true profile exhibits a maximum in the PAN mixing ratio close to the surface in the upper
panels (a and b), while the true profile peaks in the mid-troposphere in the lower panels (c and d). In each
of the profile plots, the black dashed line shows the prior, the two red lines show two different true profiles,
and the two blue lines show the retrieved profiles. In order to demonstrate the reduction in lower
tropospheric sensitivity associated with cloudy cases, panels on the right (b and d) show retrievals where a
cloud with effective optical depth of 0.7 is placed at 600 hPa (dotted line). These can be directly compared
with panels on the left (a and c), which show equivalent condition clear-sky retrievals. As discussed in
Payne et al. (2014), the TES PAN retrievals do not provide information on the vertical variation of PAN. In
all cases, the degrees of freedom for signal, or number of independent pieces of vertical information in the
retrieval, is less than 1.0. This means that the shape of the retrieved result is heavily influenced by the
shape of the prior (black dashed line), as can be seen in this figure, and the vertical distribution of PAN in



each retrieval is uncertain. Figure 1 demonstrates the limitations in sensitivity of TES PAN measurements, which provide broader spatial and temporal coverage than in situ measurements, but with a compromise on sensitivity. However, the measurements can be used to validate models, provided the averaging kernel and prior are applied to model fields before comparison with the retrievals. The averaging kernels associated

with the panels presented in Figure 1 are provided in the Supplemental Information (Figure S1).

The peak sensitivity for PAN is generally between 400 – 800 hPa (Payne et al., 2014), but a comparison between TES PAN transect observations coincident with Front Range Air Pollution and Photochemistry Éxperiment (FRAPPÉ)  observations (Figure 2) show that TES can be sensitive to PAN in the boundary layer when boundary layer PAN is extremely elevated. As an example, Figure 2 presents in

situ observations from a flight during FRAPPÉ made with a thermal dissociation chemical ionization mass spectrometer (TD-CIMS) (Zheng et al., 2011). In situ data show mixing ratios up to 2.2 ppbv were observed within the boundary layer. Afternoon mixing ratios > 1 ppbv were also observed at the Boulder Atmospheric Observatory (BAO) ground site on this day (Zaragoza et al., 2017). The overlaid TES data in Figure 2a (parallelograms) show that TES is sensitive to the elevated boundary layer values despite the

presence of high clouds (dashed line Figure 2c). Figure 2 also shows that TES has sensitivity to PAN below 800 hPa, but the retrieval places the additional PAN higher up in the atmosphere. Because of the lack of vertical information, we define the tropospheric average for a given retrieval as the average retrieved PAN between 800 hPa and the tropopause. This is what is plotted in Figure 2a and used throughout the paper.

For the analysis presented below, we use PAN observations from TES over North America in

July, from 2006 to 2009. We only include data with DOFS $\geq$ 0.6 to ensure that the retrievals are dominated by real observed information rather than the *a priori*. This conservative choice means that we are primarily basing our analysis on retrievals with high PAN. The impact of this choice can be seen when we compare the PAN distribution observed by TES under different conditions later in Section 3.2

**2.2 NOAA Hazard Mapping System (HMS) Smoke Plume Extent**

We segregate the TES PAN retrievals by whether or not the TES footprint coincides with a smoke plume identified by the NOAA Hazard Mapping System (HMS). NOAA HMS is an interactive satellite image and graphics system developed by the National Environmental Satellite, Data, and Information Service (NESDIS). Using satellite imagery, trained analysts identify the geographic extent of smoke-plumes in the atmospheric column over North America (Rolph et al., 2009; Ruminski et al., 2006). Visible-

band geostationary (~15 minute refresh rate) imagery, occasionally assisted by infrared, is used to detect smoke plumes in the atmospheric column (Ruminski et al., 2006); because smoke plumes are primarily identified with visible imagery, the analyzed smoke plume extent is only representative of local daylight hours.

Plumes are analyzed multiple times on a given day and can be nested. For this work all

overlapping plumes (either nested or analyzed at different times) are merged into a single plume. This dataset does not contain information about the vertical location or depth of smoke in the atmospheric column. As discussed in Brey et al. (2017), the number and extent of smoke plumes in this HMS dataset is



a conservative estimate. In particular, it becomes challenging to identify smoke as it dilutes during transport or mixes with anthropogenic haze. Thus our estimate of the number of PAN retrievals impacted by smoke may be a lower bound. For this work, we follow the overlap methods described in Brey et al. (2017). We matched all TES PAN retrievals based on UTC day. As discussed in Brey et al. (2017), most of the large wildfire plumes occurring in July over the western U.S. are very large and last several days. So we would expect that pairing the overnight retrievals with the plume from the prior day (i.e. matching based only on UTC day) is not likely to change our results, and that is the case. We have repeated all our calculations using only the daytime retrievals, and the choice to use all the retrievals does not change the results.

### 2.3 HYSPLIT trajectories

As part of a case study presented in Section 3.3, we use the Hybrid Single-Particle Lagrangian Integrated Trajectory (HYSPLIT) model (Draxler, 1998) (http://ready.arl.noaa.gov/HYSPLIT.php) to simulate the air mass history of a subset of TES PAN retrievals associated with relatively fresh (0 – 2 days of atmospheric processing) smoke. HYSPLIT has been used extensively to model the transport of smoke (*e.g.*, Stein et al. (2015) and Brey et al. (2017)). For this application, the HYSPLIT model is driven by global meteorological data from the Global Data Assimilation System (GDAS) archive (ftp://arlftp.arlhq.noaa.gov/pub/archives/gdas1). GDAS has a time step of 3-hours, horizontal grid spacing of $1^{\circ}$ latitude by $1^{\circ}$ longitude (~120 km), and 23 pressure surfaces between 1000 and 20 hPa (Kanamitsu, 1989). We initialized 5-day backward trajectories for set of single TES retrievals at the retrieval times and locations. In the case study in Section 3.3 we used trajectories initialized at 2, 4 and 6 km above ground level (agl). As the vertical distribution of PAN in each retrieval is uncertain (Section 2.1), we calculated backward trajectories using these three altitudes to test the sensitivity of our results to the choice of initialization altitude.

## 3 Results

### 3.1 North American TES PAN Retrievals Associated with Smoke

The first four panels of Figure 3 show the spatial distribution of TES PAN retrievals over the U.S. and southern Canada for the month of July 2006 to 2009. All retrievals plotted in this figure have DOF > 0.6. The retrievals are colored red when they fall within a NOAA HMA smoke plume. A large fraction of the TES retrievals (15-32%) during this month overlap smoke plumes; the largest percentage of retrievals associated with smoke occurred in July 2008 (32%), though this year does not display a high percentage of detection compared to other years and the average tropospheric PAN measured by TES is not larger than other years (Supplemental Figure 1). The number of major wildfires over the U.S. has large seasonal and interannual variability (Brey et al., 2017). Wildfires in summer 2008 were particularly intense over California associated with record-breaking lightning and aggravated drought. Figure 3a shows a cluster of TES PAN retrievals over California associated with this smoke. The dense smoke, which spread substantially downwind, was sampled from the NASA DC-8 aircraft as part of the Arctic Research of the Composition of the Troposphere from Aircraft and Satellites (ARCTAS-CARB) campaign (Hecobian et al., 2011; Singh et al., 2010; Singh et al., 2012), and we show this data in Section 3.3. Elevated smoke was





also observed at surface sites downwind throughout the month of July (Gyawali et al., 2009). As part of
ARCTAS-B, Alvarado et al. (2010) also documented major PAN enhancements in fresh wildfire plumes
sampled over Canada during July 2008. July 2008 was also associated with special observations from TES,
providing a relatively high number of attempted retrievals this month (red line in Supplemental Figure 2).
Figure 3f presents the seasonal transition for 2006 in smoke-plume polygon overlap from late spring (May)

to early autumn (September).  During this example year, the percentage of TES PAN retrievals with DOF >
0.6 associated with smoke peaked in July (20%), but Figure 3e suggests that this was not a notably high
percentage of smoke-impacted retrievals. A much higher percentage of DOF > 0.6 retrievals were smoke-
impacted in July 2008.

        Panels a and b of Figure 4 show the distribution of tropospheric average TES PAN in the subset of

retrievals overlapping HMS smoke plume polygons in July 2006-2009. The distributions of tropospheric
PAN in the subset of retrievals with DOF $\geq$ 0.6 is not different between the in-smoke cases (leftmost red
box plot in Figure 4a) and the not-in-smoke cases (Blue-Grey box plot in Figure 4a). The choice to only
include data with DOFS $\geq$ 0.6, pushes the median tropospheric average PAN substantially higher than
using all the available TES data. Thus the percent of retrievals impacted by smoke shown in Figure 3

reflects only situations with substantially elevated PAN in the atmospheric column. Imposing an additional
cloud optical depth filter does not substantially change the distribution of tropospheric average PAN (see
Supplemental Figure 4). The other two red distributions in Figure 4a reflect additional criteria designed to
ensure that the PAN associated with smoke in the atmospheric column exists in the free troposphere where
we expect TES to be most sensitive. We show the PAN distribution for in-smoke cases that also coincide

with TES 510hPa CO > 120 ppbv and TES 510hPa CO > 150 ppbv. There are differences between these
subsets of data and the not-in smoke cases.  As discussed further in Section 3.3, background CO in July in
the northern mid-latitudes is expected to be ~85 ppbv. Both criteria (510 hPa CO > 120 ppbv or 510 hPa
CO > 150 ppbv) represent conservative indicators of smoke in the free troposphere. The latter subset is
shown because this designation has been used previously (Alvarado et al., 2011), and we use this subset in

our calculation of enhancement ratios in Section 3.3.

        Figure 4c and 4d present the distribution of tropospheric mean CO associated with the successful
PAN measurements. There is higher CO associated with TES retrievals that overlap HMS smoke polygons
(median = 100 ppbv versus 92 ppbv for both day and night retrievals), and the upper tail of the CO
distribution includes retrievals with tropospheric average CO above 200 ppbv. The difference in CO

distributions in and out of smoke provides confidence in the use of the HMS smoke product as a smoke-
impact filter. The tropospheric average CO distributions are shown for reference because we combine
tropospheric average CO with tropospheric average PAN to calculate PAN enhancement ratios in Section
3.3.  There are several other factors that may also contribute to the patterns shown in Figure 4 that are
worth noting. In general, TES is more sensitive to CO than PAN in the lowermost atmosphere, and the

HMS smoke product, which contains no vertical information, includes smoke plumes near the surface and
higher in the column. Though the sensitivity to clouds appears to be modest in our data, the TES CO



retrievals are even less sensitive overall to the presence of cloud than the TES PAN retrievals. Third, many of the smoke-impacted TES retrievals are located substantially downwind of the source fires. PAN has a substantially shorter lifetime than CO in the warm lower atmosphere in summer.

**3.2 July 2007 Case Study**

TES observations allow measurements of smoke plumes over North America at various ages, even in the same day. Figure 5 shows the spatial distribution of TES retrievals with DOF > 0.6 over the U.S. and southern Canada for the month of July 2006 to 2009 that overlapped HMS smoke plume polygons. These points are the red colored retrieval locations in Figure 3, but here they have been colored by the day of the

month. The filled dots represent points where TES 510 hPa CO > 150 ppbv, and these are the points used to calculate PAN enhancement ratios in Section 3.3. The presence of same colored dots demonstrate that wide swaths of North America can have smoke located somewhere in the atmospheric column on a given day, and that the smoke is associated with elevated PAN (> 200 pptv) in the atmospheric column. As discussed in Brey et al. (2017), smoke plumes vary in size substantially. Small plumes cover < 100 km$^2$ and smoke

plumes from major fire complexes can spread over several Western States or entire Canadian Provinces. For example, Figure 5 shows elevated PAN both directly over and east of Hudson Bay in late July 2008 associated with fires in northern Saskatchewan.

Next we present a case study of fires in Idaho and Montana during July 2007 that connects PAN enhancements associated with HMS smoke plumes to regions impacted by fires, indicating that the TES

sensitivity is often sufficient to measure elevated PAN several days downwind of a fire. Figure 6 presents the locations of TES retrievals with elevated (DOF > 0.6) PAN on 22 and 23 July 2007, red and purple dots respectively, along with FIRMS MODIS hotspots (Giglio et al., 2006; Giglio et al., 2003) on those two dates. The TES PAN retrievals are located almost directly over active fires in Idaho on 22 July, but this does not absolutely ensure that the PAN is from fresh smoke. As discussed in *Payne et al.* [2014] TES is

most sensitive to PAN in the mid-troposphere, and we do not have injection height information for these specific fires. The TES PAN retrievals on 23 July (located over rural areas in North and South Dakota) are not located directly over active fires, but they do overlap HMS smoke polygons. The purple lines show HYSPLIT backward trajectories initialized from 4 km at the locations of the retrievals on 23 July. The trajectories show that the major fire complexes in Idaho and Montana likely contributed to the smoke

observed by TES on 23 July (purple dots). If so, this smoke was approximately 1-2 days old at the time of the retrieval. The trajectories show that the smoke observed over South Dakota is likely older (2-3 days of atmospheric ageing). We initialize the trajectories from various heights (2, 4 and 6 km) because the TES PAN retrievals offer no vertical information, and all these trajectories are plotted in Supplemental Figure S3. The smoke filled a relatively thick layer based on available CALIPSO data. A CALIPSO overpass on

23 July 2007 (lower panel of Figure 6) shows an aerosol layer identified largely as *elevated smoke* extending from the surface to ~5 km over this region.

**3.3 PAN Enhancements in North American Biomass Burning Plumes**




Figure 7 presents a histogram of PAN enhancement ratios in the subset of retrievals shown in Figure 7 (colored dots), as well as these values for the entire suite of retrievals that overlap HMS smoke

polygons and also are likely to have elevated PAN and CO in the free troposphere (TES CO > 150 hPa). PAN enhancement ratios were estimated using tropospheric average PAN and tropospheric average CO. We performed this calculation using a CO background of 80 and 90 ppbv. Background CO in the Northern Hemisphere is generally between 80 and 90 ppbv (*e.g.* Parrish et al. (1991)) with significant year-to-year variability largely driven by boreal forest fire emissions (Wotawa et al., 2001). Thus the lower mixing ratio

(80 ppbv) is closer to estimates of background CO in the Northern Hemisphere. The upper mixing ratio (90 ppbv) reflects the median tropospheric average CO (91 ppbv) in the PAN TES retrievals not overlapping HMS Smoke Polygons (blue-grey points in Figure 3). Though we repeated this calculation with various assumptions of background CO mixing ratios, this choice does not impact the major key point we draw from Figure 7. Even with our conservative CO criteria applied, the TES PAN data offer the opportunity to

calculate tropospheric average PAN enhancement relative to CO for a large number of smoke samples (N =159) over a variety of regions and distances downwind from fires. The median PAN enhancement ratio relative to CO calculated using a background PAN mixing ratio of 0.1 ppbv and a background CO mixing ratios of 90 ppbv is 0.43 %. When we assume a higher PAN background mixing ratio of 0.2 ppbv with this background CO mixing ratio, the median PAN enhancement ratio from the TES data is 0.29 %. As we

show next, these values are similar to that reported from in situ measurements.

We have not been able to identify a case study where the TES data can be used to examine the evolution of the enhancement ratio of PAN relative to CO in a plume. Restricting ourselves to the conservative criteria of 510 hPa CO > 150 ppbv severely reduces the sample size (from 1151 to 159). In addition, the 5 km x 8 km footprint of TES combined with the lack of vertical sensitivity makes it difficult

to establish the age of the smoke contributing to the enhanced PAN and CO. There could be multiple layers of smoke in the column, of various ages. Tracking plumes with aircraft allows for a more precise determination of plume age. In addition, PAN does not simply dilute proportionally to CO because its dissociation is also a function of temperature, which also depends on altitude.

We compare the TES column PAN enhancement ratios to enhancement ratios of PAN relative to

CO observed during July 2008 during the ARCTAS/CARB field campaign (Hecobian et al., 2011). Smoke identification within the aircraft dataset is discussed in detail in Hecobian et al. (2011) and not repeated here. Alvarado et al. (2010) report mean PAN enhancement ratios for boreal plumes using this same dataset. They report enhancement ratios of 0.34 ± 0.35 % (range = 0.09 % to 1.43 %) for fresh plumes and 0.28 ± 0.36 % (range = 0.16 % to 0.68 %) for old plumes. In Alvarado et al. (2010), fresh plumes were

designated as those where propene was correlated with CO, and aged plumes were designated as plumes where CO was correlated with more long-lived species, like butane, benzene and propane. The enhancement ratios were calculated using aircraft data from plume crossings using the average within-plume PAN and CO mixing ratios and assuming background mixing ratios equal to the 25[th] percentile of all measurements in the boundary layer (140 ppbv for CO and 180 pptv for PAN). To calculate enhancement



ratios presented in Figure 8, we used the 25th percentile for each trace gas for each day.  For simplicity, we
used observations at all altitudes, not just boundary layer points.  Figure 8 shows that there is a range of in
situ enhancement ratios.  Similar to the tropospheric average enhancement ratio from TES, the majority of
these enhancement ratios fall below 1%. There are retrievals with PAN enhancement ratios greater than
1%, but the number of these depends on the assumed background PAN used in the calculation. The

appropriate value to use is difficult to determine from the TES data alone, which is why a range of
estimates is presented in Figure 7. Figure 8 presents enhancement ratios calculated from in situ
measurements. This data shows that there is a higher median enhancement for plumes from fires in the
northwestern U.S., than the boreal plumes, though there are vastly different numbers of samples.

A second chance for a qualitative comparison of PAN enhancement ratios in smoke plumes is

presented in Briggs et al. (2016); summertime observations of 23 different plumes from the Mount
Bachelor Observatory indicate PAN enhancement ratios of $1.46 - 6.25$ pptv ppbv$^{-1}$ ($0.146 - 0.625$ %). This
range overlaps with the majority of the column average enhancement ratios from TES. All of the plumes
identified in Briggs et al. (2016) were from fires in northern California or southeastern and central Oregon,
so they differ from the fires intercepted during ARCTAS.

**4.0 Conclusions**

We present the first detailed analysis of TES PAN measurements over North America. Recent aircraft
observations over Colorado offer the most direct overlap of the TES PAN product with in situ aircraft
observations to date. This comparison indicates that TES can be sensitive to PAN in the boundary layer
when PAN in the boundary layer is elevated, though peak sensitivity is in the free troposphere. We use a

period with a large number of TES PAN observations ($2006 - 2009$) to investigate the contribution of fire
smoke to elevated PAN over North America in July. This type of multi-year synthesis is not possible with
any other observational dataset, and demonstrates how satellite measurements of PAN can be used to frame
new questions that cannot be answered with existing in situ measurements.

1. We segregate and examine the abundance of tropospheric average PAN relative to CO in TES retrievals

located within smoke plumes identified by the NOAA Hazard Mapping System (HMS). We find that a
large fraction of the TES retrievals (15-32%) during the month of July overlap smoke plumes during the
period $2006 - 2009$, while the largest percentage of retrievals associated with smoke occurred in July 2008
(32%). Tropospheric average CO is clearly enhanced in retrievals impacted by smoke, but a difference in
PAN between smoke-free and smoke-impacted retrievals is insignificant.

2. We compare the tropospheric average PAN enhancement relative to CO in smoke-impacted samples and
find that our satellite-based estimates largely fall within the range of enhancement ratios that have been
observed from recent aircraft and surface campaigns over western North America. While in situ
measurements represent samples from a select number of plumes, the satellite measurements offer more
samples of different plumes and observations over regions and time periods that have not been sampled by

aircraft.



3. We use a case study to illustrate that PAN enhancements associated with HMS smoke plumes can be connected to regions impacted by fires, indicating that the TES sensitivity is often sufficient to measure elevated PAN several days downwind of a fire.

4. Case studies of specific smoke events do not show a systematic pattern in PAN enhancements relative to CO as a function of distance downwind from presumed source fires. We also do not observe any consistent evolution in the PAN enhancement ratio when this calculation is done using the tropospheric maximum PAN and CO from the TES retrievals, rather than the tropospheric averages. The TES PAN data are not useful in this context because of large limitations associated with evaluating smoke age within the TES data.

PAN is considered to be the most important reservoir for $NO_x$ in the troposphere, and it plays a critical role in the redistribution of $NO_x$ to remote regions. The work presented here highlights the importance of fires as a source of PAN over North America in summer. It also shows that TES measurements of PAN can be used to complement limited in situ measurements of PAN. The apparent significant contribution of fires to elevated PAN plumes over North America underscores the importance of investigating PAN production in smoke to ultimately determine the best way to incorporate the rapid chemistry that produces PAN into chemical transport models that are used to predict background $O_3$ and exceptional $O_3$ events.

**Data Availability**: TES PAN retrievals are being processed routinely for the whole TES dataset and will be publicly available in the TES v7 Level 2 product. However, at the time of submission, the v7 processing is still underway. For netCDF files containing TES PAN data used in this study, please contact Dr. Vivienne H. Payne at Vivienne.H.Payne@jpl.nasa.gov. When the paper is accepted for final publication, we will add a text file containing the latitude, longitude, time, HMS smoke overlap status, and tropospheric average PAN and CO to the CSU digital repository ( http://hdl.handle.net/10217/180136) we have already established.

**Appendices**: N/A

**Supplemental Information:** Uploaded as a separate PDF.

**Team List and Author Contributions:**

**Emily V. Fischer** led the majority of the analysis and writing associated with this manuscript.

**Liye Zhu** provided basic statistical analyses of the TES data for the region of interest.

**Vivienne H. Payne** led the processing and development of the TES PAN data.

**John R. Worden** provided guidance on the use of the TES PAN data.

**Zhe Jiang** provided guidance on the use of the TES PAN data.

**Susan S. Kulawik** supported the algorithm development for the TES PAN retrieval.

**Steven Brey** led the overlap analysis of the TES retrievals with HMS smoke plumes.

**Arsineh Hecobian** provided the smoke designation associated with the ARCTAS aircraft data.





**Dan Gombos and Karen Cady-Pereira** performed data analysis and visualization of TES PAN distributions, concentrations, and averaging kernels from the FRAPPE aircraft and satellite data

**Frank Flocke** was responsible for the FRAPPE aircraft PAN measurements.

**Competing interests:** The authors declare that they have no conflict of interest.
**Disclaimer:** N/A

**Special issue statement:** N/A

**Acknowledgements.** This work was supported by NASA Award Number NNX14AF14G. Part of this work was carried out at the Jet Propulsion Laboratory, California Institute of Technology, under a contract with

NASA. PAN data from ARCTAS was provided by Greg Huey supported by NASA Award Number NNX08AR67G.  We thank Glenn Diskin for the use of the ARCTAS CO data.

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

            Background ozone over the United States in summer: Origin, trend, and contribution to pollution
            episodes, J. Geophy. Res., 107, ACH 11-11-ACH 11-25, 10.1029/2001JD000982, 2002.

        Fischer, E. V., Jaffe, D. A., and Weatherhead, E. C.: Free tropospheric peroxyacetyl nitrate (PAN) and
            ozone at Mount Bachelor: potential causes of variability and timescale for trend detection, Atmos.
Chem. Phys., 11, 5641-5654, 10.5194/acp-11-5641-2011, 2011.

        Fischer, E. V., Jacob, D. J., Yantosca, R. M., Sulprizio, M. P., Millet, D. B., Mao, J., Paulot, F., Singh, H.
            B., Roiger, A., Ries, L., Talbot, R. W., Dzepina, K., and Pandey Deolal, S.: Atmospheric
            peroxyacetyl nitrate (PAN): a global budget and source attribution, Atmos. Chem. Phys., 14,
            2679-2698, 10.5194/acp-14-2679-2014, 2014.

Giglio, L., Descloitres, J., Justice, C. O., and Kaufman, Y. J.: An Enhanced Contextual Fire Detection
            Algorithm for MODIS, Remote Sens. Environ., 87, 273-282, 10.1016/S0034-4257(03)00184-6,
            2003.

        Giglio, L., Csiszar, I., and Justice, C. O.: Global distribution and seasonality of active fires as observed
            with the Terra and Aqua Moderate Resolution Imaging Spectroradiometer (MODIS) sensors, J.
Geophys. Res., 111, n/a-n/a, 10.1029/2005JG000142, 2006.





Glatthor, N., von Clarmann, T., Fischer, H., Funke, B., Grabowski, U., Höpfner, M., Kellmann, S., Kiefer, M., Linden, A., Milz, M., Steck, T., and Stiller, G. P.: Global peroxyacetyl nitrate (PAN) retrieval in the upper troposphere from limb emission spectra of the Michelson Interferometer for Passive Atmospheric Sounding (MIPAS), Atmos. Chem. Phys., 7, 2775-2787, 10.5194/acp-7-2775-2007,
2007.

Gyawali, M., Arnott, W. P., Lewis, K., and Moosmüller, H.: In situ aerosol optics in Reno, NV, USA during and after the summer 2008 California wildfires and the influence of absorbing and non-absorbing organic coatings on spectral light absorption, Atmos. Chem. Phys., 9, 8007-8015, 10.5194/acp-9-8007-2009, 2009.

Hecobian, A., Liu, Z., Hennigan, C. J., Huey, L. G., Jimenez, J. L., Cubison, M. J., Vay, S., Diskin, G. S., Sachse, G. W., Wisthaler, A., Mikoviny, T., Weinheimer, A. J., Liao, J., Knapp, D. J., Wennberg, P. O., Kürten, A., Crounse, J. D., Clair, J. S., Wang, Y., and Weber, R. J.: Comparison of chemical characteristics of 495 biomass burning plumes intercepted by the NASA DC-8 aircraft during the ARCTAS/CARB-2008 field campaign, Atmos. Chem. Phys., 11, 13325-13337, 10.5194/acp-11-
13325-2011, 2011.

Hurteau, M. D., Westerling, A. L., Wiedinmyer, C., and Bryant, B. P.: Projected Effects of Climate and Development on California Wildfire Emissions through 2100, Environ. Sci. Tech., 48, 2298-2304, 10.1021/es4050133, 2014.

Jaffe, D. A., Wigder, N., Downey, N., Pfister, G., Boynard, A., and Reid, S. B.: Impact of Wildfires on
Ozone Exceptional Events in the Western U.S, Environ. Sci. Tech., 47, 11065-11072, 10.1021/es402164f, 2013.

Jiang, Z., Worden, J. R., Payne, V. H., Zhu, L., Fischer, E., Walker, T., and Jones, D. B. A.: Ozone export from East Asia: The role of PAN, J. Geophy. Res., 121, 6555-6563, 10.1002/2016JD024952, 2016.

Kanamitsu, M.: Description of the NMC Global Data Assimilation and Forecast System, Wea. Forecasting, 4, 335-342, 10.1175/1520-0434(1989)004<0335:DOTNGD>2.0.CO;2, 1989.

Kasibhatla, P. S., Levy, H., and Moxim, W. J.: Global NO x , HNO3, PAN, and NO y distributions from fossil fuel combustion emissions: A model study, J. Geophy. Res., 98, 7165-7180, 10.1029/92JD02845, 1993.

Keywood, M., Kanakidou, M., Stohl, A., Dentener, F., Grassi, G., Meyer, C. P., Torseth, K., Edwards, D., Thompson, A. M., Lohmann, U., and Burrows, J.: Fire in the Air: Biomass Burning Impacts in a Changing Climate, Critical Reviews in Environmental Science and Technology, 43, 40-83, 10.1080/10643389.2011.604248, 2013.

Lindaas, J., Farmer, D. K., Pollack, I. B., Abeleira, A., Zaragoza, J., Flocke, F. M., Roscioli, R., Herndon,
S., and Fischer, E. V.: The impact of aged wildfire smoke on ozone photochemistry in the Colorado Front Range, Atmos. Chem. Phys. Discuss., in review, doi:10.5194/acp-2017-171, 2017.





Mao, J., Paulot, F., Jacob, D. J., Cohen, R. C., Crounse, J. D., Wennberg, P. O., Keller, C. A., Hudman, R. C., Barkley, M. P., and Horowitz, L. W.: Ozone and organic nitrates over the eastern United States: Sensitivity to isoprene chemistry, J. Geophy. Res., 118, 11,256-211,268, 10.1002/jgrd.50817, 2013.


Mills, G. P., Sturges, W. T., Salmon, R. A., Bauguitte, S. J. B., Read, K. A., and Bandy, B. J.: Seasonal variation of peroxyacetylnitrate (PAN) in coastal Antarctica measured with a new instrument for the detection of sub-part per trillion mixing ratios of PAN, Atmos. Chem. Phys., 7, 4589-4599, 10.5194/acp-7-4589-2007, 2007.


Monks, P. S., Archibald, A. T., Colette, A., Cooper, O., Coyle, M., Derwent, R., Fowler, D., Granier, C., Law, K. S., Mills, G. E., Stevenson, D. S., Tarasova, O., Thouret, V., von Schneidemesser, E., Sommariva, R., Wild, O., and Williams, M. L.: Tropospheric ozone and its precursors from the urban to the global scale from air quality to short-lived climate forcer, Atmos. Chem. Phys., 15, 8889-8973, 10.5194/acp-15-8889-2015, 2015.


Moore, D. P., and Remedios, J. J.: Seasonality of Peroxyacetyl nitrate (PAN) in the upper troposphere and lower stratosphere using the MIPAS-E instrument, Atmos. Chem. Phys., 10, 6117-6128, 10.5194/acp-10-6117-2010, 2010.

Moritz, M. A., Parisien, M.-A., Batllori, E., Krawchuk, M. A., Van Dorn, J., Ganz, D. J., and Hayhoe, K.: Climate change and disruptions to global fire activity, Ecosphere, 3, 10.1890/ES11-00345.1, 2012.


Morris, G. A., Hersey, S., Thompson, A. M., Pawson, S., Nielsen, J. E., Colarco, P. R., McMillan, W. W., Stohl, A., Turquety, S., Warner, J., Johnson, B. J., Kucsera, T. L., Larko, D. E., Oltmans, S. J., and Witte, J. C.: Alaskan and Canadian forest fires exacerbate ozone pollution over Houston, Texas, on 19 and 20 July 2004, J. Geophy. Res., 111, n/a-n/a, 10.1029/2006JD007090, 2006.

Moxim, W. J., Levy, H., and Kasibhatla, P. S.: Simulated global tropospheric PAN: Its transport and


impact on NO x, J. Geophy. Res., 101, 12621-12638, 10.1029/96JD00338, 1996.

Müller, M., Anderson, B. E., Beyersdorf, A. J., Crawford, J. H., Diskin, G. S., Eichler, P., Fried, A., Keutsch, F. N., Mikoviny, T., Thornhill, K. L., Walega, J. G., Weinheimer, A. J., Yang, M., Yokelson, R. J., and Wisthaler, A.: In situ measurements and modeling of reactive trace gases in a small biomass burning plume, Atmos. Chem. Phys., 16, 3813-3824, 10.5194/acp-16-3813-2016,


2016.

Pandey Deolal, S., Henne, S., Ries, L., Gilge, S., Weers, U., Steinbacher, M., Staehelin, J., and Peter, T.: Analysis of elevated springtime levels of Peroxyacetyl nitrate (PAN) at the high Alpine research sites Jungfraujoch and Zugspitze, Atmos. Chem. Phys., 14, 12553-12571, 10.5194/acp-14-12553-2014, 2014.


Parrish, D. D., Trainer, M., Buhr, M. P., Watkins, B. A., and Fehsenfeld, F. C.: Carbon monoxide concentrations and their relation to concentrations of total reactive oxidized nitrogen at two rural U.S. sites, J. Geophy. Res., 96, 9309-9320, 10.1029/91JD00047, 1991.



Payne, V. H., Alvarado, M. J., Cady-Pereira, K. E., Worden, J. R., Kulawik, S. S., and Fischer, E. V.:
 Satellite observations of peroxyacetyl nitrate from the Aura Tropospheric Emission Spectrometer,
515  Atmos. Meas. Tech., 7, 3737-3749, 10.5194/amt-7-3737-2014, 2014.

Payne, V. H., Fischer, E. V., Worden, J. R., Jiang, Z., Zhu, L., Kurosu, T. P., and Kulawik, S. S.: Spatial
 variability in tropospheric peroxyacetyl nitrate in the tropics from infrared satellite observations in
 2005 and 2006, Atmos. Chem. Phys. Discuss., 2016, 1-21, 10.5194/acp-2016-1047, 2016.

Pfister, G. G., Wiedinmyer, C., and Emmons, L. K.: Impacts of the fall 2007 California wildfires on surface
520  ozone: Integrating local observations with global model simulations, Geophys. Res. Lett., 35, n/a-
 n/a, 10.1029/2008GL034747, 2008.

Pinder, R. W., Gilliland, A. B., and Dennis, R. L.: Environmental impact of atmospheric NH3 emissions
 under present and future conditions in the eastern United States, Geophys. Res. Lett., 35, n/a-n/a,
 10.1029/2008GL033732, 2008.

525 Rastigejev, Y., Park, R., Brenner, M. P., and Jacob, D. J.: Resolving intercontinental pollution plumes in
 global models of atmospheric transport, J. Geophy. Res., 115, n/a-n/a, 10.1029/2009JD012568,
 2010.

Rolph, G. D., Draxler, R. R., Stein, A. F., Taylor, A., Ruminski, M. G., Kondragunta, S., Zeng, J., Huang,
 H.-C., Manikin, G., McQueen, J. T., and Davidson, P. M.: Description and Verification of the
530  NOAA Smoke Forecasting System: The 2007 Fire Season, Wea. Forecasting, 24, 361-378,
 10.1175/2008WAF2222165.1, 2009.

Ruminski, M., Kondragunta, S., Draxler, R. R., and Zheng, W.: Recent changes to the Hazard mapping
 System, 15th International Emission Inventory Conference: Reinventing Inventories, New Ideas in
 New Orleans, New Orleans, LA, 2006,

535 Scholze, M., Knorr, W., Arnell, N. W., and Prentice, I. C.: A climate-change risk analysis for world
 ecosystems, PNAS, 103, 13116-13120, 2006.

Singh, H. B., and Hanst, P. L.: Peroxyacetyl nitrate (PAN) in the unpolluted atmosphere: An important
 reservoir for nitrogen oxides, Geophys. Res. Lett., 8, 941-944, 10.1029/GL008i008p00941, 1981.

Singh, H. B.: Reactive nitrogen in the troposphere, Environ. Sci. Tech., 21, 320-327, 10.1021/es00158a001,
540  1987.

Singh, H. B., Anderson, B. E., Brune, W. H., Cai, C., Cohen, R. C., Crawford, J. H., Cubison, M. J., Czech,
 E. P., Emmons, L., Fuelberg, H. E., Huey, G., Jacob, D. J., Jimenez, J. L., Kaduwela, A., Kondo,
 Y., Mao, J., Olson, J. R., Sachse, G. W., Vay, S. A., Weinheimer, A., Wennberg, P. O., and
 Wisthaler, A.: Pollution influences on atmospheric composition and chemistry at high northern
545  latitudes: Boreal and California forest fire emissions, Atmos. Environ., 44, 4553-4564,
 10.1016/j.atmosenv.2010.08.026, 2010.

Singh, H. B., Cai, C., Kaduwela, A., Weinheimer, A., and Wisthaler, A.: Interactions of fire emissions and
 urban pollution over California: Ozone formation and air quality simulations, Atmos. Environ., 56,
 45-51, 10.1016/j.atmosenv.2012.03.046, 2012.





Stein, A. F., Draxler, R. R., Rolph, G. D., Stunder, B. J. B., Cohen, M. D., and Ngan, F.: NOAA's
            HYSPLIT Atmospheric Transport and Dispersion Modeling System, B. Am. Meteorol. Soc., 96,
            2059-2077, 10.1175/BAMS-D-14-00110.1, 2015.

       Tanimoto, H., Furutani, H., Kato, S., Matsumoto, J., Makide, Y., and Akimoto, H.: Seasonal cycles of
            ozone and oxidized nitrogen species in northeast Asia 1. Impact of regional climatology and
photochemistry observed during RISOTTO 1999–2000, J. Geophy. Res., 107, ACH 6-1-ACH 6-
            20, 10.1029/2001JD001496, 2002.

       Travis, K. R., Jacob, D. J., Fisher, J. A., Kim, P. S., Marais, E. A., Zhu, L., Yu, K., Miller, C. C., Yantosca,
            R. M., Sulprizio, M. P., Thompson, A. M., Wennberg, P. O., Crounse, J. D., St. Clair, J. M.,
            Cohen, R. C., Laughner, J. L., Dibb, J. E., Hall, S. R., Ullmann, K., Wolfe, G. M., Pollack, I. B.,
Peischl, J., Neuman, J. A., and Zhou, X.: Why do models overestimate surface ozone in the
            Southeast United States?, Atmos. Chem. Phys., 16, 13561-13577, 10.5194/acp-16-13561-2016,
            2016.

       Val Martin, M., Heald, C. L., Lamarque, J. F., Tilmes, S., Emmons, L. K., and Schichtel, B. A.: How
            emissions, climate, and land use change will impact mid-century air quality over the United
States: a focus on effects at national parks, Atmos. Chem. Phys., 15, 2805-2823, 10.5194/acp-15-
            2805-2015, 2015.

       Wang, Y., Jacob, D. J., and Logan, J. A.: Global simulation of tropospheric O3-NO x -hydrocarbon
            chemistry: 3. Origin of tropospheric ozone and effects of nonmethane hydrocarbons, J. Geophy.
            Res., 103, 10757-10767, 10.1029/98JD00156, 1998.

Westerling, A. L., Hidalgo, H. G., Cayan, D. R., and Swetnam, T. W.: Warming and Earlier Spring
            Increase Western U.S. Forest Wildfire Activity, Science, 313, 940-943, 2006.

       Westerling, A. L.: Increasing western US forest wildfire activity: sensitivity to changes in the timing of
            spring, Philos. Trans. R. Soc. B, 371, 2016.

       Wiegele, A., Glatthor, N., Höpfner, M., Grabowski, U., Kellmann, S., Linden, A., Stiller, G., and von
Clarmann, T.: Global distributions of C2H6, C2H2, HCN, and PAN retrieved from MIPAS
            reduced spectral resolution measurements, Atmos. Meas. Tech., 5, 723-734, 10.5194/amt-5-723-
            2012, 2012.

       Wotawa, G., Novelli, P. C., Trainer, M., and Granier, C.: Inter-annual variability of summertime CO
            concentrations in the Northern Hemisphere explained by boreal forest fires in North America and
Russia, Geophys. Res. Lett., 28, 4575-4578, 10.1029/2001GL013686, 2001.

       Yates, E. L., Iraci, L. T., Singh, H. B., Tanaka, T., Roby, M. C., Hamill, P., Clements, C. B., Lareau, N.,
            Contezac, J., Blake, D. R., Simpson, I. J., Wisthaler, A., Mikoviny, T., Diskin, G. S., Beyersdorf,
            A. J., Choi, Y., Ryerson, T. B., Jimenez, J. L., Campuzano-Jost, P., Loewenstein, M., and Gore,
            W.: Airborne measurements and emission estimates of greenhouse gases and other trace
constituents from the 2013 California Yosemite Rim wildfire, Atmos. Environ., 127, 293-302,
            10.1016/j.atmosenv.2015.12.038, 2016.



Yue, X., Mickley, L. J., Logan, J. A., and Kaplan, J. O.: Ensemble projections of wildfire activity and carbonaceous aerosol concentrations over the western United States in the mid-21st century, Atmos. Environ., 77, 767-780, 10.1016/j.atmosenv.2013.06.003, 2013.

Zaragoza, J., Callahan, S., E. E. McDuffie, J. Kirkland, P. Brophy, L. Durrett, D. K. Farmer, Y. Zhou, B. Sive, F. Flocke, G. Pfister, C. Knote, A. Tevlin, J. Murphy, and E. V. Fischer, Observations of acyl peroxy nitrates during the Front Range Air Pollution and Photochemistry Experiment (FRAPPE), J. Geophys. Res., 122, 10.1002/2017JD027337, 2017.

Zhang, L., Jacob, D. J., Downey, N. V., Wood, D. A., Blewitt, D., Carouge, C. C., van Donkelaar, A.,

Jones, D. B. A., Murray, L. T., and Wang, Y.: Improved estimate of the policy-relevant background ozone in the United States using the GEOS-Chem global model with $1/2° \times 2/3°$ horizontal resolution over North America, Atmos. Environ., 45, 6769-6776, 10.1016/j.atmosenv.2011.07.054, 2011.

Zheng, W., Flocke, F. M., Tyndall, G. S., Swanson, A., Orlando, J. J., Roberts, J. M., Huey, L. G., and

Tanner, D. J.: Characterization of a thermal decomposition chemical ionization mass spectrometer for the measurement of peroxy acyl nitrates (PANs) in the atmosphere, Atmos. Chem. Phys., 11, 6529-6547, 10.5194/acp-11-6529-2011, 2011.

Zhu, L., Fischer, E. V., Payne, V. H., Worden, J. R., and Jiang, Z.: TES observations of the interannual variability of PAN over Northern Eurasia and the relationship to springtime fires, Geophys. Res.

Lett., 42, 7230-7237, 10.1002/2015GL065328, 2015.

Zhu, L., Fischer, E. V., Payne, V. H., Walker, T., Worden, J. R., Jiang, Z., and Kulawik, S. S.: PAN in the Eastern Pacific Free Troposphere: A Satellite View of the Sources, Seasonality, Interannual Variability and Timeline for Trend Detection, J. Geophy. Res., 122, 10.1002/2016JD025868, 2017.


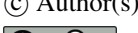



**Figures**

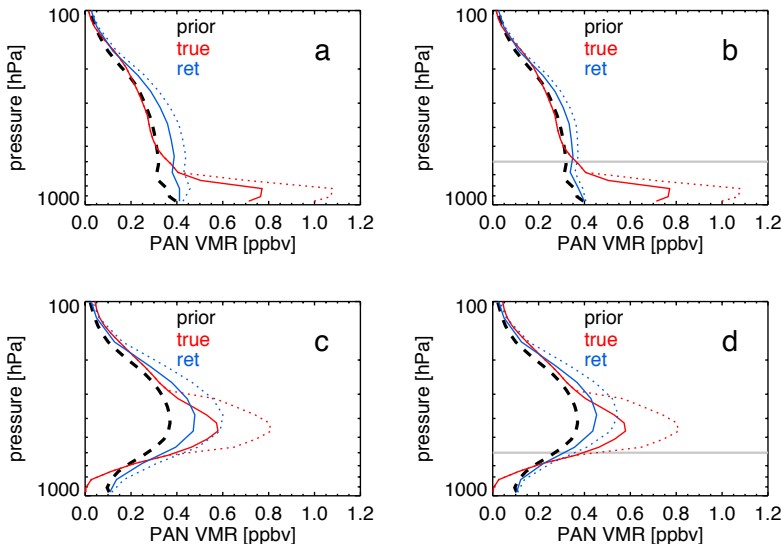

**Figure 1**: Simulated TES PAN retrievals for four different hypothetical conditions where the black dashed
line shows the prior, the two red lines show two different true profiles, and the two blue lines show the
retrieved profiles. The true profile exhibits a maximum in the vmr close to the surface in the upper panels
(a and b), while the true profile peaks in the mid-troposphere in the lower panels (c and d). Panels on the
left (a and c) show clear-sky retrievals while panels on the right (b and d) show retrievals where a cloud

with effective optical depth of 0.7 is placed at 600 hPa (dotted line). Corresponding averaging kernels are
provided in the Supplementary Information.





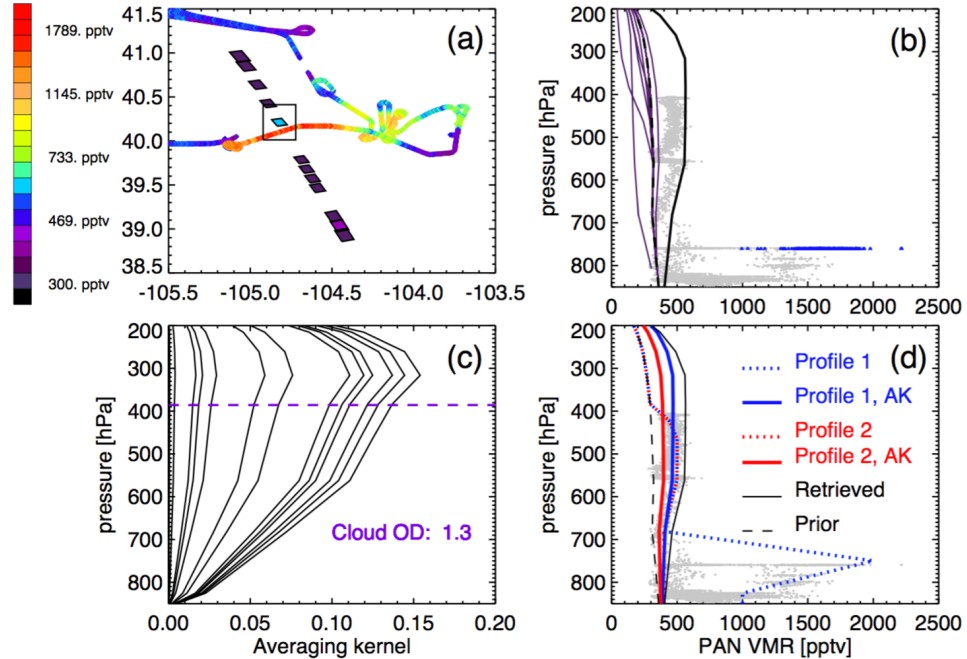

**Figure 2**: a) Map showing FRAPPÉ aircraft and TES tropospheric average satellite observations of PAN over the Colorado Front Range on 29 July 2014. TES data show elevated PAN near the location where the aircraft data show highest values for that day. b) All aircraft observations for 29 July 2014 are shown in grey. Blue points show aircraft data within 0.1° longitude and 0.2° latitude of the most elevated TES PAN observation. TES retrieved PAN profiles for 29 July 2014 are also shown. The elevated case is shown by the solid black line, while other cases are shown in purple solid lines. The black dashed line shows the TES a priori profile used in these retrievals. c) TES averaging kernels for this case. The retrieval indicates that a high cloud is present, with optical depth 1.3, leading to reduced sensitivity below the cloud. d) The blue dotted line shows a profile constructed to approximate the aircraft measurements, where PAN is highly elevated in the lower atmosphere. The blue solid line shows this same profile after smoothing with the TES prior and averaging kernel matrix for this scene. The red dotted line shows a hypothetical profile with no enhancement below 680 hPa, while the red solid line shows that same profile smoothed with the TES prior and averaging kernel. The difference between the red and the blue solid lines indicates that TES has sensitivity to the boundary layer enhancement in this case.





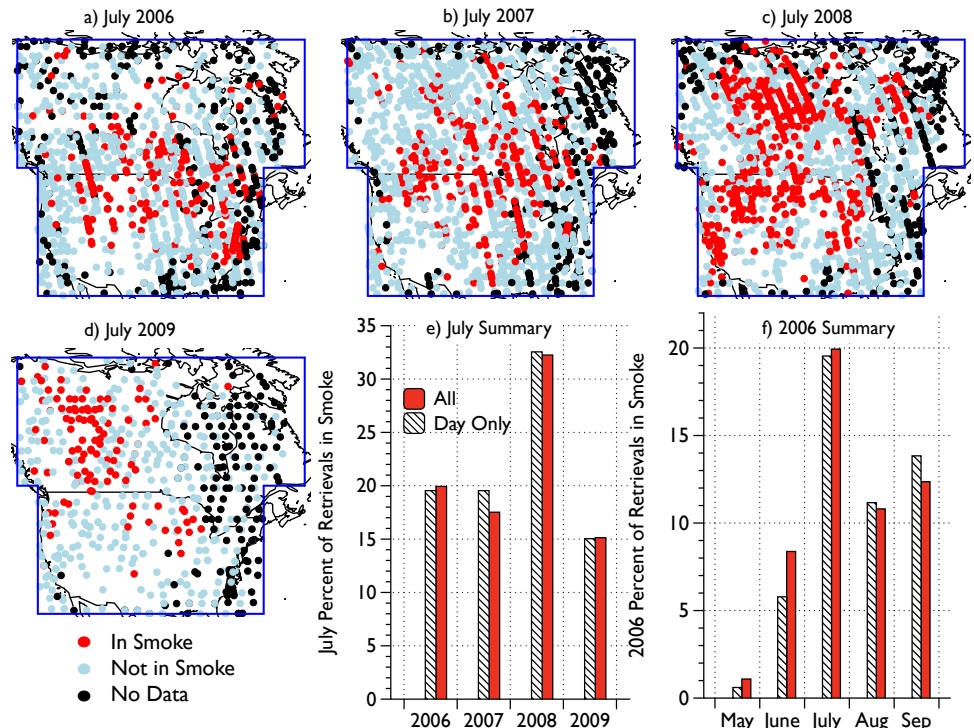


**Figure 3:** Panels a) through d): PAN TES retrievals with DOF> 0.6 co-located with NOAA Hazard
Mapping System smoke polygons (red), and PAN TES retrievals with DOF > 0.6 not co-located with
NOAA Hazard Mapping System smoke polygons (grey). The black dots indicate PAN TES retrievals with
DOF> 0.6 during times with no NOAA HMS data. The blue lines surround the regions included in the

calculations in Figures 2 and 4: $125^o$ W - $70^o$ W, $30^o$ N – $50^o$ N and $130^o$ W - $60^o$ W, $50^o$ N – $70^o$ N.  e)
Percent of TES PAN retrievals overlapping HMS smoke plume polygons for July 2006 – 2009. f) Percent
of TES PAN retrievals overlapping HMS smoke plume polygons for May – September 2006. In panels e)
and f) the red bars indicate the percentage of all retrievals overlapping smoke plumes, and the striped bars
indicate the percentage of daytime retrievals overlapping smoke plumes. Pairing was done using the

matching UTC day.





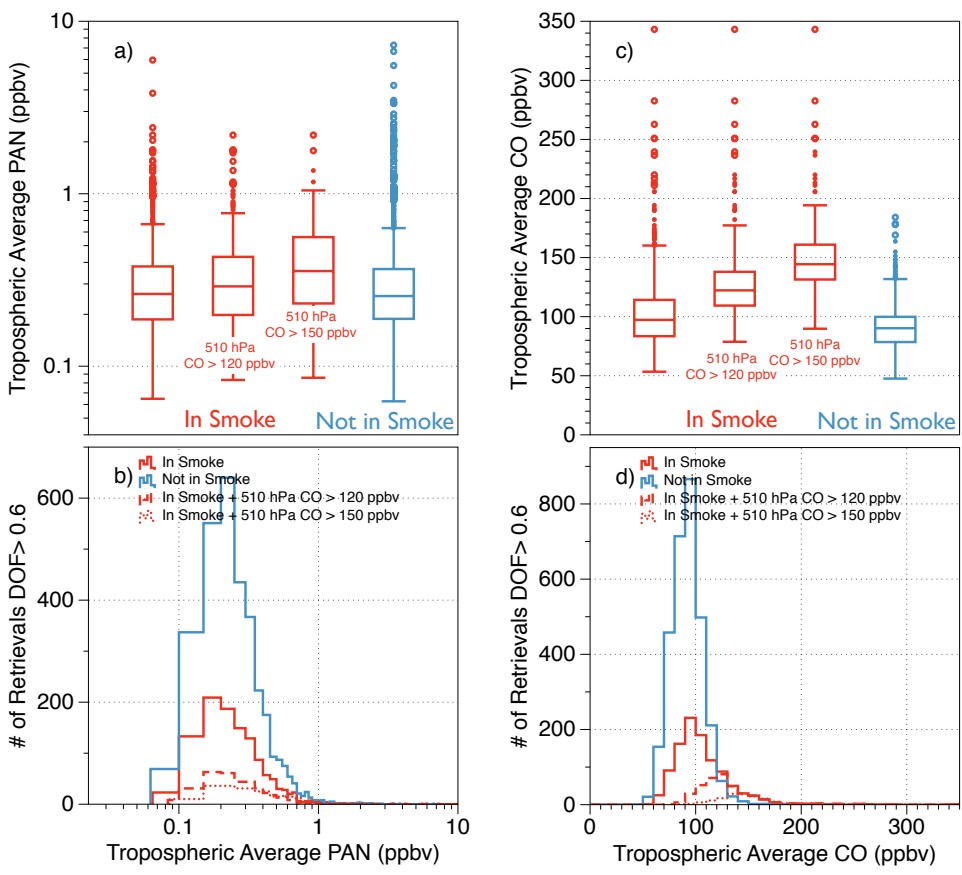


**Figure 4:** a) Box plots of July 2006 – 2009 North American TES PAN retrievals overlapping HMS smoke
plume polygons ("In Smoke"; red; N =1151), TES PAN retrievals not overlapping HMS smoke plume
polygons ("Not In Smoke"; blue-grey; N = 2917), and TES PAN retrievals that overlap HMS smoke
plumes and coincide with 510 hPa CO greater than either 120 ppbv (N = 255) or 150 ppbv (N = 139). b)
Histograms of July 2006 – 2009 TES PAN retrievals segregated as in a). c) Box plots of July 2006 – 2009
TES CO retrievals coincident with the TES PAN retrievals segregated as in a). d) Histograms of July 2006
– 2009 TES CO retrievals coincident with the TES PAN retrievals segregated as in a). The box plots
display the interquartile range for each subset and the dots represent outliers.




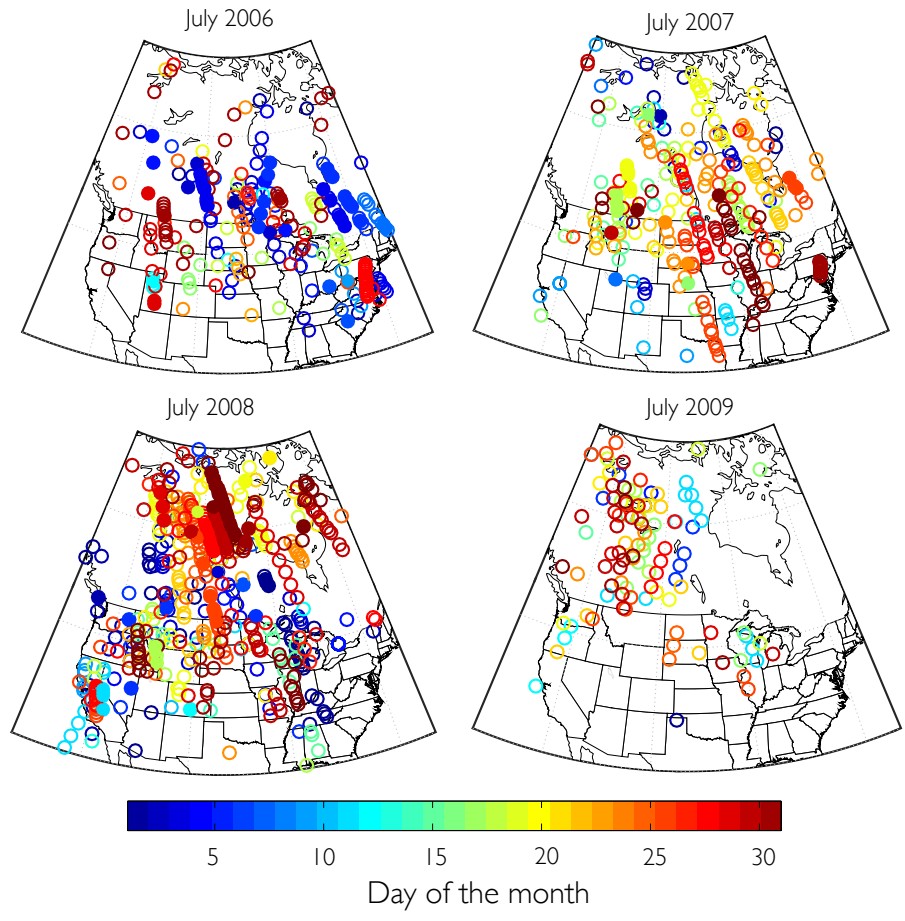


**Figure 5:** Successful TES PAN retrievals overlapping NOAA HMS smoke polygons for July 2006 to July 2009 colored by the day of the month. Filled circles denote the set of retrievals that also coincide with 510 hPa CO greater than 150 ppbv. This set of point is used to calculated PAN enhancement ratios relative to CO in Figure 7.






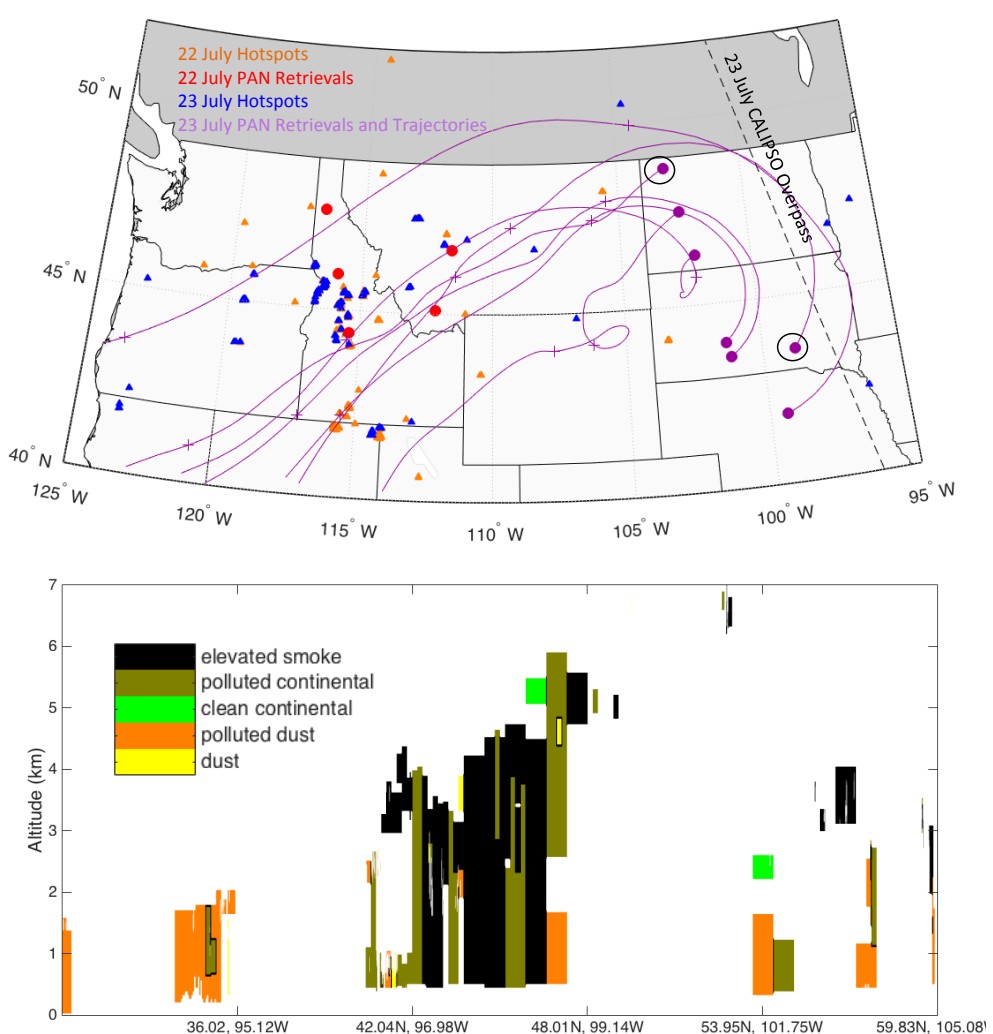

**Figure 6:** Top panel: Case study of TES PAN retrievals overlapping HMS smoke polygons 22 – 23 July 2007. Orange triangles represent FIRMS MODIS Hotspots for 22 July (Product MCD14ML; https://firms.modaps.eosdis.nasa.gov/download/request.php). Blue triangles represent FIRMS MODIS Hotspots for 23 July. Red circles indicate TES PAN retrievals on 22 July, and purple circles represent TES PAN retrievals on 23 July. We have circled the two retrievals in this set with 510 hPa CO greater than 150 ppbv. The PAN enhancement ratios for these points are noted in Figure 7. The purple lines signify 5 day HYSPLIT backward trajectories initialized at each TES retrieval at 4 km. The purple '+' signifies 24 hours of transport time on the 4 km trajectories. The black dashed line shows the location of the CALIPSO swath shown in the lower panel. Lower panel: CALIPSO aerosol subtype observed on 23 July 2007. CALIPSO Science Team (2016), CALIPSO/CALIOP Level 2, Vertical Feature Mask Data, version 4.10, Hampton,



VA, USA: NASA Atmospheric Science Data Center (ASDC), Accessed by Emily V. Fischer at doi:

10.5067/CALIOP/CALIPSO/LID_L2_VFM-Standard-V4-10






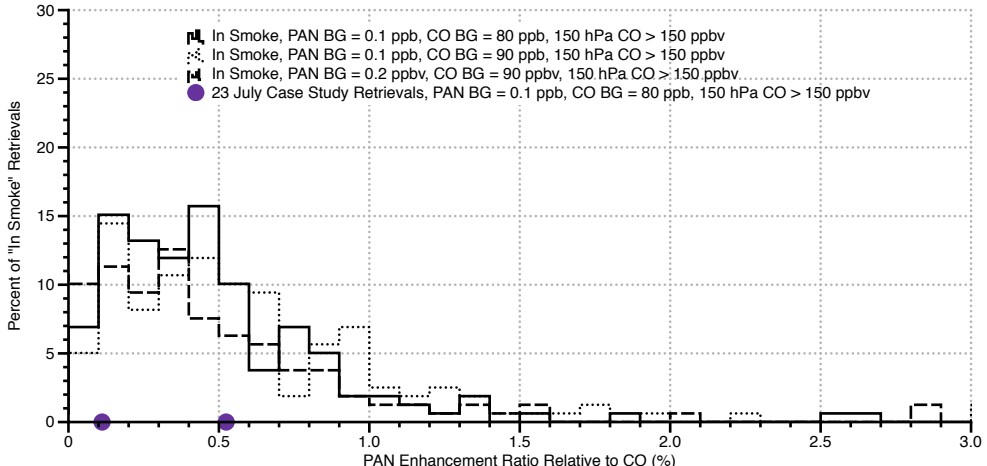

**Figure 7:** Histogram of estimated PAN enhancement ratios based on tropospheric mean PAN and CO from July 2006 – 2009 North American TES PAN retrievals overlapping HMS smoke plume polygons. The solid

red line represents enhancement ratios calculated using an assumed PAN background of 0.1 ppbv with an assumed CO background of 80 ppbv. The dotted red line represents enhancement ratios calculated using an assumed PAN background of 0.1 ppbv with an assumed CO background of 90 ppbv. The red dots are the enhancement ratios for the 5 retrievals on 22 July 2007 plotted in Figure 5 associated with fresh smoke. The purple dots are the enhancement ratios for the 6 retrievals on 23 July 2007 plotted in Figure 5

associated with transported smoke. These specific enhancement ratios were calculated using an assumed CO background of 80 ppbv, similar to the solid red line. The dashed orange line represents enhancement ratios calculated using a significantly higher assumed PAN background of 0.2 ppbv with an assumed CO background of 90 ppbv. In all cases, negative values are not shown.



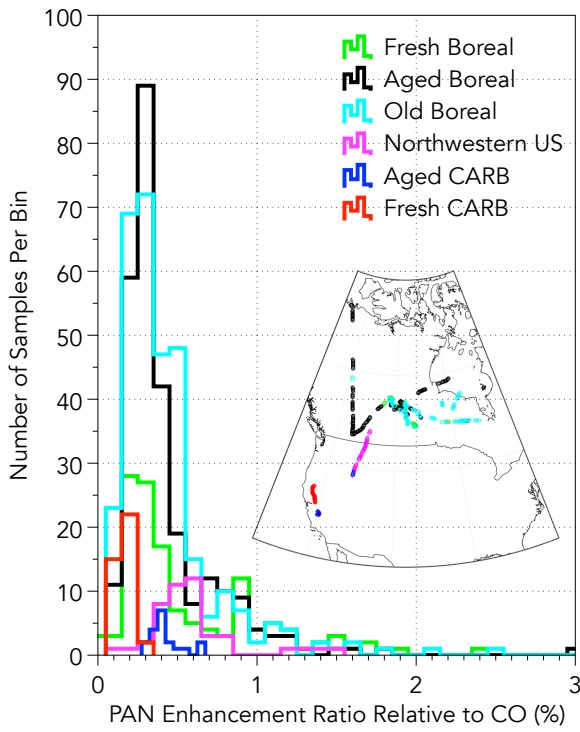


**Figure 8:** Histogram of estimated PAN enhancement ratios based on in situ measurements of fire plumes described in Hecobian et al. (2011). Enhancement ratios were calculated using the 25[th] percentile for each trace gas during the corresponding flight day. These ratios were calculated using the 1-minute merged data.
