# Peer review of "Using TES Retrievals to Investigate PAN in North American Biomass Burning Plumes"

_Atmospheric Chemistry and Physics, 2017_

## Referee Comment (RC1) · Anonymous Referee #1 · 16 Nov 2017

The manuscript by Fischer et al., presents tropospheric measurements of satellite retrieved peroxyacetyl nitrate (PAN) over North America and investigates the changes in concentrations linked to fires. Overall, the manuscript would be a nice addition to the existing literature (e.g. Payne et al., 2014, 2016) as there are limited flight campaigns measuring PAN and MIPAS can only retrieve it in the UTLS. The compositing of TES PAN retrievals under smoke plumes is also an interesting way to investigate potential enhancements of PAN related to fires. Therefore, once the comments below are addressed, this manuscript should be accepted for publication in ACP.

Major comments:

[Figure]

As TES is the only satellite currently measuring lower tropospheric PAN (to the best of my knowledge anyway), it would be useful to see the spatial distribution of PAN at different tropospheric levels. In neither this paper nor the Payne et al., (2014, 2016) manuscripts, there are very few spatial maps of TES PAN. In Payne et al., (2016), Figure 1a shows a noisy spatial distribution of TES PAN in the tropics. Therefore, it would be useful if this study could add another figure (e.g. between Figure 1 and Figure 2) showing the PAN distribution over N. America (e.g. the July 2006-2009 average on a regular grid instead of individual retrievals on several tropospheric levels) highlighting the average PAN hotspots and potential outflow of PAN from source regions.

The presentation of the manuscript needs to be improved as several of the Figures have been mislabelled in the text and it is difficult to follow. In Figure 7, there is reference to red lines, but all the lines are grey/black, again making it difficult to read the paper.

Section 3.3 needs to be made clearer as discussion of the PAN:CO ratios is rather rushed. For instance, adding some equations into Section 3.3 on how the enhancement ratios are calculated would be useful. Again, as Figure 7 has misleading colours, it is difficult to work out what the authors are trying to say in this section.

Minor comments:

P1 L68: Would be good to reference of Ungermann et al., (2016) who investigate PAN in the summer-time Asian monsoon region using Earth observation measurements. On Line 62-64, the authors states "much of our understanding of the distribution of PAN outside urban areas rests on data from aircraft missions interpreted with global chemical transport models". I think it would be useful to reference a few papers that have utilised CTMs and satellite data to investigate PAN (e.g. Fadnavis et al., 2014; Pope et al., 2016).

P2 L102: What do the authors mean by "True profiles"? In Figure 1, would the true profile by the retrieved profile? P2 L106-107: "As discussed in Payne et al. (2014),

the TES PAN retrievals do not provide information on the vertical variation of PAN". This does not make sense. PAN is retrieved at several vertical levels and the AKs will provide information on the vertical sensitivity.

P2 L107-109: IF a DOF < 1 means a retrieval is heavily influenced by the aprioi then why do the authors often use the criteria of the DOF > 0.6?

P3 L128-129: The authors state TES has sensitivity to enhanced PBL PAN, but the concentrations are much lower than that of the aircraft?

P3 L127-128: Add tropospheric column definition to the Figure 2 caption.

P4 L 153-154: Please explain "i.e. matching based only on UTC day" more clearly.

P4 L178: Do the authors mean Supplementary Figure (SF) 2 not SF1? Also, why is the red axis (number of attempts) over the Pacific Ocean? This needs to be explained more clearly?

P4 L180: Figure 3c instead of 3a?

P7 L 259: Coloured dots? I can only see purple dots.

Figure 2b: Why is there such a large discrepancy between aircraft (blue) and TES (black) PAN?

Figure 2d: Worth adding equation in main text or caption how the AKs are applied.

Figure 2d: The difference between the red and blue solid lines looks tiny, so how does this show TES has good sensitivity?

Figure 6: Useful to add a CALIPSO track line to the top panel map...i.e. where did CALIPSO cross the domain?

Figure 7: Where are the red lines/dots?

Figure 8: State that the data is from ARCTAS.

Figure S2: How do the authors define "elevated PAN"?

References:

Fadnavis, S., Schultz, M. G., Semeniuk, K., Mahajan, A. S., Pozzoli, L., Sonbawne, S., Ghude, S. D., Kiefer, M., and Eckert, E.: Trends in peroxyacetyl nitrate (PAN) in the upper troposphere and lower stratosphere over southern Asia during the summer monsoon season: regional impacts, Atmos. Chem. Phys., 14, 12725–12743, doi:10.5194/acp-14-12725-2014, 2014.

Payne, V. H., Alvarado, M. J., Cady-Pereira, K. E., Worden, J. R., Kulawik, S. S., and Fischer, E. V.: Satellite observations of peroxyacetyl nitrate from the Aura Tropospheric Emission Spectrometer, Atmos. Meas. Tech., 7, 3737-3749, 10.5194/amt-7-3737-2014, 2014.

Payne, V. H., Fischer, E. V., Worden, J. R., Jiang, Z., Zhu, L., Kurosu, T. P., and Kulawik, S. S.: Spatial variability in tropospheric peroxyacetyl nitrate in the tropics from infrared satellite observations in 2005 and 2006, Atmos. Chem. Phys. Discuss., 2016, 121, 10.5194/acp-2016-1047, 2016.

Pope, R. J., Richards, N. A. D., Chipperfield, M. P., Moore, D. P., Monks, S. A., Arnold, S. R., Glatthor, N., Kiefer, M., Breider, T. J., Harrison, J. J., Remedios, J. J., Warneke, C., Roberts, J. M., Diskin, G. S., Huey, L. G., Wisthaler, A., Apel, E. C., Bernath, P. F., and Feng, W.: Intercomparison and evaluation of satellite peroxyacetyl nitrate observations in the up- per troposphere–lower stratosphere, Atmos. Chem. Phys., 16, 13541–13559, doi:10.5194/acp-16-13541-2016, 2016.

Ungermann, J., Ern, M., Kaufmann, M., Müller, R., Spang, R., Ploeger, F., Vogel, B., and Riese, M.: Observations of PAN and its confinement in the Asian summer monsoon anticyclone in high spatial resolution, Atmos. Chem. Phys., 16, 8389–8403, doi:10.5194/acp-16-8389-2016, 2016.

[Figure]

2017.

---

## Referee Comment (RC2) · Anonymous Referee #2 · 21 Dec 2017

In this manuscript, the authors investigate the overlap of TES peroxyacyl nitrate (PAN) detections (defined as DOF>0.6) with HMS smoke extent and TES CO retrievals in the western United States. The authors 1) quantify the fraction of "enhanced" TES retrievals (DOF>0.6) overlapping HMS smoke extent by month and year, 2) perform two case studies using FRAPPE, HYSPLIT, MODIS and CALIPSO data and 3) evaluate the ratio of TES tropospheric average CO and PAN enhancements.

Based on feedback from preliminary reviews, the authors have removed comparison of TES PAN measurements with model simulations from the manuscript but have not substantially improved the quantitative characterization of the retrieved product. Additionally, there are editorial and typographical errors in the manuscript (e.g., Figure 6 top panel appears almost identical to Figure S3; seemingly erroneous in text figure references – Line 178).

Further characterization of TES PAN retrievals is important and the authors have worked towards that goal. However, I recommend that the authors use language that more precisely conveys the uncertainty of the product. In the introduction, for example, the authors state that "Satellite measurements [i.e., TES] are essential to understand the seasonal cycle and interannual variations of PAN (L65)." Here and similarly throughout, use of a strong word like "Essential" connotes long-standing maturity and widespread use. In this case, I would recommend using a "potential tool" instead of "essential". While this is one specific example, edits should be made throughout the manuscript to address this concern.

The examples below indicate where the authors can add information to the manuscript from their already accomplished analyses that will be useful to the reader.

–Figure SI 4 seems to imply that there is no statistically significant difference between "In smoke" and "Not in smoke" retrievals. Is this expected? This needs to be addressed in the main text.

–One of the paper's main findings is that ∼15-32% of PAN detections (DOF > 0.6) overlap with HMS smoke extent. Is that a larger or smaller number than expected? Please report the percent of all attempted TES retrievals that overlap HMS smoke extent for context.

–The authors report that there is no statistical difference between PAN "detections" (DOF > 0.6) that do and do not overlap HMS smoke extent. Please discuss this finding in context of past airborne or mountain-top in situ studies: should we expect there to be a statistical difference in PAN concentrations in biomass burning influenced air and other polluted airmasses.

–There is one case study involving CALIPSO data. The paper might benefit from a more statistically robust analysis of CALIPSO and TES data. The meridional offset of ∼500 km between the two sensor tracks (Figure 6) could likely be overcome by the incorporation of reanalysis wind products and the potentially higher quality smoke information provided by CALIPSO.

L92 - "The TES PAN retrievals shown here were processed using a prototype algorithm for the area and time periods of interest." Please provide more details regarding differences between this prototype and the operational retrievals. What is the source of a priori profiles in the retrieval? What is the tropospheric average PAN concentration in the a priori profiles? The prior profile shown in Figure panel 2d shows PAN concentrations between 300 ppt and 400 ppt – if so, please justify use of an assumed "background" concentration of 100 or 200 ppt in the PAN/CO enhancement analysis in Section 3.3.

L96 – "For footprints where the spectra show strong evidence of this silicate feature in the surface emissivity ( this can occur over rocky or sandy surfaces ) TES PAN retrievals are not attempted." What fraction of retrievals is discarded by this requirement? Does dust aerosol have a similar silicate absorption or emission feature that should be considered in the retrievals?

L118 – What is "extremely elevated"? Please quantify.

L130 – "We only include data with DOFS > 0.6 to ensure that the retrievals are dominated by real observed information." "Dominated" is a rather strong word to describe DOFs > 0.6. The meaning of "real observed information" is unclear.

L132 - "This conservative choice means that we are primarily basing our analysis on retrievals with high PAN" What is the mean and standard deviation of retrieved PAN concentrations?

L172 – 193 – The purpose of this paragraph is not clear to me. Furthermore the

evidence is not all that convincing as the expected relationships between smoke and TES PAN detections is not consistent or as expected. I did not find Figures 3 or 5 to be particularly helpful either.

L194-210 + Figure 4 – The data in this paragraph and figure can be used to compute a PAN:CO ratio that is less dependent on assumptions of background contributions. Based on a cursory visual analysis of Figure 4, a value of 0.3% PAN:CO appears reasonable.

Figure 4 – The data that is "not in smoke" should be separated similarly to the data that is "in smoke." Is the relationship between CO and PAN similar "not in smoke" retrievals similar to their relationship "in smoke" (see above comment)? Should that result be expected based on previous comparisons of anthropogenic (not in smoke enhancement) and biomass burning influenced (in smoke) air?

Editorial comments:

Below is a non-exhaustive list of editorial suggestions or errors. I recommend that the authors thoroughly proofread before re-submission.

L1 – Re-define PAN as peroxyacyl nitrate in the body of the manuscript.

L140 – "infrared" [imagery].

L174 "NOAA HMA smoke plume"

L178 – "Supplemental Figure 1" – Perhaps intended for Supplemental Figure 2

Figure S3 and Figure 6 top panel appear to be identical.

---

## Author Comment (AC1) · 13 Feb 2018

The manuscript by Fischer et al., presents tropospheric measurements of satellite retrieved peroxyacetyl nitrate (PAN) over North America and investigates the changes in concentrations linked to fires. Overall, the manuscript would be a nice addition to the existing literature (e.g. Payne et al., 2014, 2016) as there are limited flight campaigns measuring PAN and MIPAS can only retrieve it in the UTLS. The compositing of TES PAN retrievals under smoke plumes is also an interesting way to investigate potential enhancements of PAN related to fires. Therefore, once the comments below are addressed, this manuscript should be accepted for publication in ACP.

Major comments:

As TES is the only satellite currently measuring lower tropospheric PAN (to the best of my knowledge anyway), it would be useful to see the spatial distribution of PAN at different tropospheric levels. In neither this paper nor the Payne et al., (2014, 2016) manuscripts, there are very few spatial maps of TES PAN. In Payne et al., (2016), Figure 1a shows a noisy spatial distribution of TES PAN in the tropics. Therefore, it would be useful if this study could add another figure (e.g. between Figure 1 and Figure 2) showing the PAN distribution over N. America (e.g. the July 2006-2009 average on a regular grid instead of individual retrievals on several tropospheric levels) highlighting the average PAN hotspots and potential outflow of PAN from source regions.

*We agree that showing a spatial distribution is interesting. Exactly as suggested, we have added the July 2006-2009 average on a regular 2x2 degree grid. However, we have only done this as a tropospheric average. As discussed in Payne et al. (2014) and also in Section 2.1, the TES PAN retrievals do not provide information on the vertical variation of PAN. In all cases, the degrees of freedom for signal, or number of independent pieces of vertical information in the retrieval, is less than 1.0. To address the true spirit of this suggestion we have also added a panel that shows all the data points individually; this allows readers to see the noise in the data. We also include the entire region of retrievals that were processed so that readers can view the distribution off-land as well. We have added another paragraph describing this new Figure (now Figure 2 as suggested) in section 2.1.*

*"Figure 2 shows the July 2006 – 2009 tropospheric average PAN. When all the existing TES data is gridded, there are several large patterns that emerge. 1) Average tropospheric PAN mixing ratios in the TES observations generally increase with latitude during the month of July over North America. 2) Average tropospheric PAN mixing ratios generally decrease from west to east. 3) As can be seen in later figures, there are relatively few retrievals per grid box over the southwestern U.S. Though there are relatively few samples (~5-20 per 2x2°grid box), relatively high mixing ratios (0.6 ppbv) are observed over the Colorado Front Range.*

The presentation of the manuscript needs to be improved as several of the Figures have been mislabelled in the text and it is difficult to follow. In Figure 7, there is reference to red lines, but all the lines are grey/black, again making it difficult to read the paper.

*We apologize for the labels on the original Figure 7. We changed the lines from red to black and color-coordinated the dots with the original Figure 6, and then it looks like only half of the caption for the original Figure 7 was updated. We have fixed this.*

Section 3.3 needs to be made clearer as discussion of the PAN:CO ratios is rather rushed. For instance, adding some equations into Section 3.3 on how the enhancement ratios are calculated would be useful. Again, as Figure 7 has misleading colours, it is difficult to work out what the authors are trying to say in this section.

*We sincerely believe that this section is now much easier to understand with the caption for Figure 7 (now Figure 8) corrected. As indicated in the comment above and noted by the second reviewer, we mislabeled black lines as red in the submitted caption. There was also an incorrect reference to Figure 7 (now Figure 8) in Section 3.3, which should have pointed readers to Figure 6 (now Figure 7). Now that these typos*

*have been fixed, this section should be much easier to follow. However, as suggested by the reviewer, we have also added an equation describing the calculation of the PAN enhancement ratios, and several sentences at the start of the section that point readers to a reference that discusses enhancement ratios and their pitfalls (Yokelson et al., 2013). There are two key points to this section, and we now state both of them in the introductory paragraph. 1) The tropospheric PAN enhancement ratios from TES fall within the range of relevant aircraft measurements over North America. 2) There are many pitfalls associated with using enhancement ratios as observed from TES to study the evolution of PAN in the smoke plumes we have identified here.*

Minor comments:
P1 L68: Would be good to reference of Ungermann et al., (2016) who investigate PAN in the summer-time Asian monsoon region using Earth observation measurements. On Line 62-64, the authors states "much of our understanding of the distribution of PAN outside urban areas rests on data from aircraft missions interpreted with global chemical transport models". I think it would be useful to reference a few papers that have utilised CTMs and satellite data to investigate PAN (e.g. Fadnavis et al., 2014; Pope et al., 2016).

*Thank you very much for pointing out these newer references. We have added citations to all of them in the suggested locations.*

P2 L102: What do the authors mean by "True profiles"? In Figure 1, would the true profile by the retrieved profile?

*The "true profile" is the actual atmospheric profile. We have updated the caption and text to indicate this.*

P2 L106-107: "As discussed in Payne et al. (2014), the TES PAN retrievals do not provide information on the vertical variation of PAN". This does not make sense. PAN is retrieved at several vertical levels and the AKs will provide information on the vertical sensitivity. P2 L107-109: IF a DOF < 1 means a retrieval is heavily influenced by the aprioi then why do the authors often use the criteria of the DOF > 0.6?

*This threshold value of DOF > 0.6 was chosen to be consistent with a signal to noise ratio (SNR) greater than 1 (Payne et al. 2014), and this criteria has been used in all the papers that have presented TES PAN data thus far. We have added these sentences to the manuscript to clarify this choice. It is also worth noting that the shape of the retrieved profile is always heavily influenced by the shape of the a priori profile for these measurements (see response to next comment).*

P3 L128-129: The authors state TES has sensitivity to enhanced PBL PAN, but the concentrations are much lower than that of the aircraft?

*For nadir retrievals of molecules with weak spectral signatures where the DOFS <1.0, the shape of the retrieved profile is heavily influenced by the shape of the prior. Since the prior profile for this case peaks in the mid-troposphere, the retrieved profile will also peak in the mid-tropopshere. A large enhancement in boundary layer PAN shows up in the TES radiances as a small enhancement in the PAN signal. A small enhancement in the mid-troposphere would also show up in the TES radiances as a small enhancement in the PAN signal. The nature of the measurement is such that it is not possible to distinguish between these two scenarios in the TES radiances. An example is provided in Figure 2 in Payne et al. (2014). Therefore, although we demonstrate for this case that TES has some sensitivity to elevated PAN in the boundary layer, for the more general case where we do not have co-located in-situ profile measurements, we would only be able to say that there is some enhancement in PAN somewhere in the column.*

*We have added this discussion to the second to last paragraph of section 2.1. This now reads:*

*"The peak sensitivity for PAN is generally between 400 – 800 hPa (Payne et al., 2014), but a comparison between TES PAN transect observations coincident with Front Range Air Pollution and Photochemistry Éxperiment (FRAPPÉ) observations (Figure 2) show that TES can have some degree of sensitivity to PAN in the boundary layer when boundary layer PAN is elevated. As an example, Figure 3 presents in situ observations from a flight during FRAPPÉ made with a thermal dissociation chemical ionization mass*

*spectrometer (TD-CIMS) (Zheng et al., 2011). Mean PAN observed by the C-130 below 3 km during the field campaign was 481 pptv (Zaragoza et al., 2017). This particular day (29 July) was one of the four days identified by Zaragoza et al. (2017) with the highest surface PAN mixing ratios observed at the Boulder Atmospheric Observatory. The overlaid TES data in Figure 3a (parallelograms) show an enhancement in the TES PAN (as shown by the TES observation highlighted by a black square) in the vicinity of aircraft measurements of highly elevated PAN values in the boundary layer indicating that in this case TES is weakly sensitive to the elevated boundary layer values despite the presence of high clouds (dashed line Figure 3c). Figure 3 also shows red and blue lines corresponding to application of the averaging kernel for this case to hypothetical "true" profiles with and without the enhancement in the boundary layer. The red and blue lines show that TES has some sensitivity to PAN below 800 hPa, but the retrieval places the additional PAN higher up in the atmosphere. While the difference between the red and the blue solid lines in Figure 3d is small, it is non-zero indicating that TES has some sensitivity to the boundary layer enhancement in this case."*

P3 L127-128: Add tropospheric column definition to the Figure 2 caption.

*This information was added. Figure 2 is now Figure 3.*

P4 L 153-154: Please explain "i.e. matching based only on UTC day" more clearly.

*Matching by UTC day is explained in the following sentences, but we added on additional one for clarification. This now reads: "We matched all TES PAN retrievals based on UTC day. This means that overnight retrievals are paired with the plume from the prior day. As discussed in Brey et al. (2017), most of the large wildfire plumes occurring in July over the western U.S. are very large and last several days. So we would expect that pairing the overnight retrievals with the plume from the prior day (i.e. matching based only on UTC day) is not likely to change our results, and that to be the case. We have repeated all our calculations using only the daytime retrievals, and the choice to use all the retrievals does not change the results."*

P4 L178: Do the authors mean Supplementary Figure (SF) 2 not SF1? Also, why is the red axis (number of attempts) over the Pacific Ocean? This needs to be explained more clearly?

*Yes, we mean Supplementary Figure 2. This typo has been corrected. The second comment is also the product of a typo in the caption for Supplementary Figure 2. This sentence is supposed to point readers to a comparable figure in Zhu et al. (2017), but that reference is missing. Similar data for the Pacific Ocean is presented there for the same set of months. This has been fixed.*

P4 L180: Figure 3c instead of 3a?

*Yes, this should refer to 3c instead of 3a. This has been corrected.*

P7 L 259: Coloured dots? I can only see purple dots.

*Yes, this should say purple to be less confusing. This has been fixed by adding a more specific sentence.*

*"Figure 8 presents a histogram of PAN enhancement ratios in the subset of retrievals that overlap HMS smoke polygons and also are likely to have elevated PAN and CO in the free troposphere (TES CO > 150 hPa). The purple dots designate the two retrievals shown in Figure 7 that meet these strict criteria."*

Figure 2b: Why is there such a large discrepancy between aircraft (blue) and TES (black) PAN?

*As discussed above, for nadir retrievals of molecules with weak spectral signatures where the DOFS <1.0, the shape of the retrieved profile is heavily influenced by the shape of the prior. Since the prior profile for this case peaks in the mid-troposphere, the retrieved profile will also peak in the mid-tropopshere. A large enhancement in boundary layer PAN shows up in the TES radiances as a small enhancement in the PAN*

*signal. A small enhancement in the mid-troposphere would also show up in the TES radiances as a small enhancement in the PAN signal. The nature of the measurement is such that it is not possible to distinguish between these two scenarios in the TES radiances. An example is provided in Figure 2 in Payne et al. [2014]. Therefore, although we demonstrate for this case that TES has some sensitivity to elevated PAN in the boundary layer, for the more general case where we do not have co-located in-situ profile measurements, we would only be able to say that there is some enhancement in PAN somewhere in the column.*

*As discussed above, we have added more text to this section. We are not claiming good sensitivity to the boundary layer. This example provides the first direct evidence of any sensitivity to PAN in the boundary layer for TES.*

Figure 2d: Worth adding equation in main text or caption how the AKs are applied.

*We have added substantial additional text and equations to Section 2.1 to address this comment.*

Figure 2d: The difference between the red and blue solid lines looks tiny, so how does this show TES has good sensitivity?

*We have not tried to claim "good" sensitivity. Rather this example shows that TES has some sensitivity. To make this clear, we have again added substantially more details to the latter part of Section 2.1, in particular, the updated paragraph now reads:*

*"The peak sensitivity for PAN is generally between 400 – 800 hPa (Payne et al., 2014), but a comparison between TES PAN transect observations coincident with Front Range Air Pollution and Photochemistry Éxperiment (FRAPPÉ) observations (Figure 2) show that TES can have some degree of sensitivity to PAN in the boundary layer when boundary layer PAN is elevated. As an example, Figure 3 presents in situ observations from a flight during FRAPPÉ made with a thermal dissociation chemical ionization mass spectrometer (TD-CIMS) (Zheng et al., 2011). Mean PAN observed by the C-130 below 3 km during the field campaign was 481 pptv (Zaragoza et al., 2017). This particular day (29 July) was one of the four days identified by Zaragoza et al. (2017) with the highest surface PAN mixing ratios observed at the Boulder Atmospheric Observatory. The overlaid TES data in Figure 3a (parallelograms) show an enhancement in the TES PAN (as shown by the TES observation highlighted by a black square) in the vicinity of aircraft measurements of highly elevated PAN values in the boundary layer indicating that in this case TES is weakly sensitive to the elevated boundary layer values despite the presence of high clouds (dashed line Figure 3c). Figure 3 also shows red and blue lines corresponding to application of the averaging kernel for this case to hypothetical "true" profiles with and without the enhancement in the boundary layer. The red and blue lines show that TES has some sensitivity to PAN below 800 hPa, but the retrieval places the additional PAN higher up in the atmosphere. While the difference between the red and the blue solid lines in Figure 3d is small, it is non-zero indicating that TES has some sensitivity to the boundary layer enhancement in this case."*

Figure 6: Useful to add a CALIPSO track line to the top panel map. . .i.e. where did CALIPSO cross the domain?

*This was included in the figure already as a dashed line labeled "CALIPSO Overpass".*

Figure 7: Where are the red lines/dots?

*We apologize for the labels on the original Figure 7. We changed the lines from red to black and color-coordinated the dots with the original Figure 6, and then it looks like only half of the caption for the original Figure 7 was updated. We have fixed this.*

Figure 8: State that the data is from ARCTAS.

*As suggested, we have changed the first sentence of the caption to also directly reference ARCTAS and not just Hecobian et al. (2011). This reads "Histogram of estimated PAN enhancement ratios based on in situ measurements of fire plumes described in Hecobian et al. (2011) from the ARCTAS campaign."*

Figure S2: How do the authors define "elevated PAN"?

*We have removed this wording because it is confusing. TES has a high detection limit, and that is already stated in the methods. This word was not needed here.*

References:

Fadnavis, S., Schultz, M. G., Semeniuk, K., Mahajan, A. S., Pozzoli, L., Sonbawne, S., Ghude, S. D., Kiefer, M., and Eckert, E.: Trends in peroxyacetyl nitrate (PAN) in the upper troposphere and lower stratosphere over southern Asia during the summer monsoon season: regional impacts, Atmos. Chem. Phys., 14, 12725–12743, doi:10.5194/acp-14-12725-2014, 2014.

Payne, V. H., Alvarado, M. J., Cady-Pereira, K. E., Worden, J. R., Kulawik, S. S., and Fischer, E. V.: Satellite observations of peroxyacetyl nitrate from the Aura Tropospheric Emission Spectrometer, Atmos. Meas. Tech., 7, 3737-3749, 10.5194/amt-7-3737- 2014, 2014.

Payne, V. H., Fischer, E. V., Worden, J. R., Jiang, Z., Zhu, L., Kurosu, T. P., and Kulawik, S. S.: Spatial variability in tropospheric peroxyacetyl nitrate in the tropics from infrared satellite observations in 2005 and 2006, Atmos. Chem. Phys. Discuss., 2016, 121, 10.5194/acp-2016-1047, 2016.

Pope, R. J., Richards, N. A. D., Chipperfield, M. P., Moore, D. P., Monks, S. A., Arnold, S. R., Glatthor, N., Kiefer, M., Breider, T. J., Harrison, J. J., Remedios, J. J., Warneke, C., Roberts, J. M., Diskin, G. S., Huey, L. G., Wisthaler, A., Apel, E. C., Bernath, P. F., and Feng, W.: Intercomparison and evaluation of satellite peroxyacetyl nitrate observations in the up- per troposphere–lower stratosphere, Atmos. Chem. Phys., 16, 13541–13559, doi:10.5194/acp-16-13541-2016, 2016.

Ungermann, J., Ern, M., Kaufmann, M., Müller, R., Spang, R., Ploeger, F., Vogel, B., and Riese, M.: Observations of PAN and its confinement in the Asian summer monsoon anticyclone in high spatial resolu

*All suggested references have been added.*

---

## Author Comment (AC2) · 13 Feb 2018

In this manuscript, the authors investigate the overlap of TES peroxyacyl nitrate (PAN) detections (defined as DOF>0.6) with HMS smoke extent and TES CO retrievals in the western United States. The authors 1) quantify the fraction of "enhanced" TES retrievals (DOF>0.6) overlapping HMS smoke extent by month and year, 2) perform two case studies using FRAPPE, HYSPLIT, MODIS and CALIPSO data and 3) evaluate the ratio of TES tropospheric average CO and PAN enhancements.

Based on feedback from preliminary reviews, the authors have removed comparison of TES PAN measurements with model simulations from the manuscript but have not substantially improved the quantitative characterization of the retrieved product.

*We respectfully disagree with the reviewer on this point. The re-submitted version included an additional figure (Figure 1) that demonstrates the limitations in the sensitivity of TES PAN measurements.*

Additionally, there are editorial and typographical errors in the manuscript (e.g., Figure 6 top panel appears almost identical to Figure S3; seemingly erroneous in text figure references – Line 178).

*We have corrected the caption for Figure 7. As discussed in response to Reviewer 1, there was a mix up in reference to "black" versus "red lines" from the earlier version. Figure S3 has not been changed as we intend it to be very similar to the top panel of Figure 7 (originally Figure 6). However, S3 shows additional trajectories than the version in the main manuscript.*

Further characterization of TES PAN retrievals is important and the authors have worked towards that goal.

*This comment seems in contrast to the one above from this reviewer, and we assume that it refers to the addition of Figure 1, which demonstrates the limitations in sensitivity of the TES PAN measurements. Perhaps the earlier comment was written before noticing Figure 1.*

However, I recommend that the authors use language that more precisely conveys the uncertainty of the product. In the introduction, for example, the authors state that "Satellite measurements [i.e., TES] are essential to understand the seasonal cycle and interannual variations of PAN (L65)."

Here and similarly throughout, use of a strong word like "Essential" connotes long-standing maturity and widespread use. In this case, I would recommend using a "potential tool" instead of "essential". While this is one specific example, edits should be made throughout the manuscript to address this concern.

*We have changed the wording here as suggested and have re-read the manuscript to remove other similar instances, but the existing set of aircraft data is insufficient to identify and understand the seasonal cycle and interannual variations in PAN in the free troposphere. The one exception is the very recently collected AToM observations, and this mission has not yet ended. The specific wording has been adjusted to read:*

*"Given the limited set of long-term in situ measurements, satellite measurements are a potential tool that can be used to investigate the seasonal cycle and interannual variability of PAN in the troposphere along with which processes contribute to these features."*

The examples below indicate where the authors can add information to the manuscript from their already accomplished analyses that will be useful to the reader.

*In addition to the specific edits listed below, we have substantially increased the information in Section 2.1 in response to some suggestions by Reviewer 1. These edits add quite a bit more information about the nature of the measurements.*

Figure SI 4 seems to imply that there is no statistically significant difference between "In smoke" and "Not in smoke" retrievals. Is this expected? This needs to be addressed in the main text.

*This is already addressed directly in the main text in Section 3.1, but we have added a few more sentences here. This result is not surprising because the HMS smoke polygons only indicate that there is smoke in the*

*column. They do not indicate whether the smoke is in the free troposphere (i.e. TES can detect it) or primarily in the boundary layer, where TES has weak sensitivity. When we restrict the TES data to only those retrievals that also have elevated CO in the free troposphere (TES 510hPa CO > 120 ppbv or TES 510hPa CO > 150 ppbv), we do see a difference between the "in smoke" and "not in smoke" retrievals.*

One of the paper's main findings is that ~15-32% of PAN detections (DOF > 0.6) overlap with HMS smoke extent. Is that a larger or smaller number than expected? Please report the percent of all attempted TES retrievals that overlap HMS smoke extent for context.

*We have added this information to Section 3.1. This now reads: "Of all the retrievals attempted in July 2006 to July 2009, the percent associated with smoke is 18%. We expect a higher fraction of overlap in the subset of data with DOF > 0.6. This threshold value of DOF > 0.6 is consistent with a signal to noise ratio (SNR) greater than 1 (Payne et al. 2014), and this subset of data only reflects conditions with elevated PAN in the atmospheric column."*

The authors report that there is no statistical difference between PAN "detections" (DOF > 0.6) that do and do not overlap HMS smoke extent. Please discuss this finding in context of past airborne or mountain-top in situ studies: should we expect there to be a statistical difference in PAN concentrations in biomass burning influenced air and other polluted airmasses.

*See our response above. HMS smoke polygons only indicate that there is smoke in the column. They do not indicate whether the smoke is in the free troposphere (i.e. TES can detect it) or primarily in the boundary layer, where TES has weak sensitivity. When we restrict the TES data to only those retrievals that also have elevated CO in the free troposphere (TES 510hPa CO > 120 ppbv or TES 510hPa CO > 150 ppbv), we do see a difference between the "in smoke" and "not in smoke" retrievals. We have added several additional references to point readers to aircraft and surface measurements of PAN enhancements in biomass burning plumes. However, we note that we use aircraft data in the paper already (Figure 9). Without an enhancement in PAN above the background, our calculations would not be meaningful. Elevated PAN is often observed in biomass burning plumes.*

There is one case study involving CALIPSO data. The paper might benefit from a more statistically robust analysis of CALIPSO and TES data. The meridional offset of ~500 km between the two sensor tracks (Figure 6) could likely be overcome by the incorporation of reanalysis wind products and the potentially higher quality smoke information provided by CALIPSO.

*It would be interesting to do a comparison between TES and CALIPSO data, but that is beyond the scope of this paper, and would need to be focused carefully. The point of Figure 6 (now Figure 7) is to show that the smoke was likely located in the lowest 5 km of the atmosphere. We also examined CALIPSO Aerosol data from an overpass located further west on 23 July 2007. There is polluted aerosol in the column during this overpass as well. The southern portions of this overpass are located west of the active fires on this day. Similar to the image that we show in Figure 6 (now Figure 7), smoke or polluted smoke is identified from the surface to 4-5 km. The swath shown in Figure 6 overlaps an HMS smoke polygon that extends from the WA/ID border to Minnesota. We also note that the trajectories displayed in Figure 6 are based on reanalysis wind products.*

[Figure]

*CALIPSO aerosol subtype observed on 23 July 2007, 10:23:44.5. CALIPSO Science Team (2016), CALIPSO/CALIOP Level 2, Vertical Feature Mask Data, version 4.10, Hampton, VA, USA: NASA Atmospheric Science Data Center (ASDC), Accessed by Emily V. Fischer at doi: 10.5067/CALIOP/CALIPSO/LID_L2_VFM-Standard-V4-10*

"The TES PAN retrievals shown here were processed using a prototype algorithm for the area and time periods of interest." Please provide more details regarding differences between this prototype and the operational retrievals. What is the source of a priori profiles in the retrieval? What is the tropospheric average PAN concentration in the a priori profiles? The prior profile shown in Figure panel 2d shows PAN concentrations between 300 ppt and 400 ppt – if so, please justify use of an assumed "background" concentration of 100 or 200 ppt in the PAN/CO enhancement analysis in Section 3.3.

*The prototype algorithm used here is effectively identical to the algorithm that has been implemented in the v7 routine Level 2 processing. We have now added further information to this effect in the text. Background PAN or CO refers to that not strongly impacted by smoke. It does not refer to the a priori.*

"For footprints where the spectra show strong evidence of this silicate feature in the surface emissivity ( this can occur over rocky or sandy surfaces ) TES PAN retrievals are not attempted." What fraction of retrievals is discarded by this requirement? Does dust aerosol have a similar silicate absorption or emission feature that should be considered in the retrievals?

*In early testing of the algorithm, we had found that desert and rocky surfaces would tend to show high initial chi-squared values for the PAN retrieval, with residual features that could only be brought within the noise by fitting strongly negative PAN VMR values. Further explanation of this issue, with a figure, is shown in Payne et al. (2014). Aside from negative values being unphysical, the current algorithm retrieves in terms of ln(vmr), so negative PAN values are not possible to retrieve within the current framework. In general, we had found that cases with high initial chi-squared values would tend to fail, and we had chosen to set a threshold (initial chi-squared > 3.0) above which retrievals are not attempted. Of course, surface emissivity features are not the only reason why there might be a high initial chi-suqared value. Another reason might be poor fits to interfering species, such as water vapor. In the current version of the algorithm, we do not attempt to explicitly track all the different reasons for failure of quality control, but instead have implemented a master quality flag. We have added the following text to address the reviewer's question about the fraction of retrievals discarded:*

*"Of the 28149 TES footprints processed for this work that fell over land, 3608 of them failed quality control. Concentrated regions of failed quality control show up as white patches in Figure 2(b). These regions are largely desert or mountainous regions."*

*The reviewer raises an interesting point about dust aerosol. There could indeed be silicate features in dust aerosol. A number of groups have looked at dust signatures in spaceborne thermal infrared radiance measurements and at their impact on other retrieval products. If the dust absorption were strong enough, this could be another reason why the TES PAN retrieval might not be attempted due to a high initial chi-squared value. If the dust absorption were sufficiently weak, this would cause the TES PAN retrieval to be biased low. A TES dust flag or dust product does not currently exist, and a rigorous assessment of the impact of dust aerosol on the TES trace gas products is outside the scope of this study, but would be worthy of consideration for future work. We have added two sentences in the manuscript on the possibility of impact from dust aerosol, with example references, and thank the reviewer for raising this point.*

L118 – What is "extremely elevated"? Please quantify.
We have removed the words "extremely elevated", and this now reads:

*"Mean PAN observed by the C-130 below 3 km during the field campaign was 481 pptv (Zaragoza et al., 2017). This particular day (29 July) was one of the four days identified by Zaragoza et al. (2017) with the highest surface PAN mixing ratios observed at the Boulder Atmospheric Observatory."*

L130 – "We only include data with DOFS > 0.6 to ensure that the retrievals are dominated by real observed information." "Dominated" is a rather strong word to describe DOFs > 0.6. The meaning of "real observed information" is unclear.

*In response to the other reviewer, we have replaced this sentence with more specific information. This now reads:*

*"We only include data with DOFS > 0.6. More specifically, this threshold value of DOF > 0.6 was chosen to be consistent with a signal to noise ratio (SNR) greater than 1 (Payne et al., 2014), and this criteria has been used in all the papers that have presented TES PAN data thus far (Zhu et al., 2015;Payne et al., 2016;Jiang et al., 2016;Zhu et al., 2017)."*

L132 - "This conservative choice means that we are primarily basing our analysis on retrievals with high PAN" What is the mean and standard deviation of retrieved PAN concentrations?

*We have added the following sentence: "The mean (standard deviation) of the retrieved tropospheric average PAN mixing ratios for DOFS > 0.6 for the region shown in the figures presented here (125$^o$ W - 70$^o$ W, 30$^o$ N – 50$^o$ N and 130$^o$ W - 65$^o$ W, 50$^o$ N – 70$^o$ N) is 0.551 (0.925) ppbv." However, this can be seen in the supplemental figures.*

L172 – 193 – The purpose of this paragraph is not clear to me. Furthermore the paper evidence is not all that convincing as the expected relationships between smoke and TES PAN detections is not consistent or as expected. I did not find Figures 3 or 5 to be particularly helpful either.

*The point of this paragraph is to show the spatial distribution of which TES retrievals overlap HMS smoke plumes each month. We have decided to keep all these figures in the main body of the manuscript. Figure 4 (now Figure 5) shows the distribution of tropospheric average PAN and CO as measured by TES within smoke plumes over the U.S. Figure 5 shows which day of the month overlaps a smoke plume. These figures are essential for the reader to understand the data, and the other reviewer did not make any negative comments about these figures.*

L194-210 + Figure 4 – The data in this paragraph and figure can be used to compute a PAN:CO ratio that is less dependent on assumptions of background contributions. Based on a cursory visual analysis of Figure 4, a value of 0.3% PAN:CO appears reasonable. Figure 4 – The data that is "not in smoke" should be separated similarly to the data that is "in smoke." Is the relationship between CO and PAN similar "not in smoke" retrievals similar to their relationship "in smoke" (see above comment)? Should that result be expected based on previous comparisons of anthropogenic (not in smoke enhancement) and biomass burning influenced (in smoke) air?

*Sure, this would be another approach, albeit a bit coarse. This quick visual approach does yield a value that agrees with many of our samples (see Figure 8 histogram). However, giving one value like this would not show the range of PAN enhancements that are observed. It does not make sense to present a PAN enhancement ratio relative to CO in non-smoke impacted samples. In these samples, the PAN and CO could have different sources. We do not know if the other samples are anthropogenically influenced. The PAN could also have been produced by lightning $NO_x$. Or the PAN and CO could be at different levels of the atmosphere. This is why we have been very conservative, presenting enhancement ratios only using the small subset of data where 510 hPa CO is great than 150 ppbv.*

Editorial comments: Below is a non-exhaustive list of editorial suggestions or errors. I recommend that the authors thoroughly proofread before re-submission.
L1 – Re-define PAN as peroxyacyl nitrate in the body of the manuscript.

*PAN is commonly known by its misnomer peroxyacetyl nitrate. This is a very minor point, and this name is used in many other manuscripts. However, we have changed the name as requested.*

L140 – "infrared" [imagery]. L174 "NOAA HMA smoke plume"

*Suggested change has been made.*

L178 – "Supplemental Figure 1" – Perhaps intended for Supplemental Figure 2

*Yes, this was also noted by Reviewer 1, and it has been corrected.*

Figure S3 and Figure 6 top panel appear to be identical.

*Figure S3 has not been changed as we intend it to be very similar to the top panel of Figure 7 (originally Figure 6). However, S3 shows additional trajectories than the version in the main manuscript. S3 looks a bit too cluttered for the main body of the paper, but it provides more information.*

---

## Author Response (AR2)

*We thank the reviewer for taking the time to re-read the manuscript a second time. We have easily made all of the suggested edits, as they were all quite minor. Our responses are in italics below each specific comment.*

The title and abstract of this manuscript suggest that one is able to identify and quantify the influence of biomass burning emissions on PAN distributions using TES v7 retrievals. The body of the text did not convince me that this is the case. The abstract and the title need to better reflect the findings of the paper.

Please consider changing the title to something more along the lines of "Are PAN enhancements in NA biomass plumes detectable in TES retrievals?"

*If it is not too late in the process, we agree that adding TES to the title is an excellent suggestion, since the focus is on TES data rather than aircraft observations. We support the following shorter title, rather than a question:*

*Using TES retrievals to investigate PAN in North American Biomass Burning Plumes*

Abstract:

"We find that 15 – 32 % of cases where elevated PAN is identified in TES observations (retrievals with DOF > 0.6) overlap smoke plumes" – Please add text defining what the range represents, and add the finding that 18% of all retrievals overlap HMS smoke extent.

*We have added this information. These sentences in the abstract now read:*

*"Depending on the year, 15 – 32 % of cases where elevated PAN is identified in TES observations (retrievals with DOF > 0.6) overlap smoke plumes during July. Of all the retrievals attempted in July 2006 to July 2009, the percent associated with smoke is 18%."*

"Using aircraft observations from the Colorado Front Range, we demonstrate that TES can be sensitive to elevated PAN in the boundary layer even in the presence of clouds." – indicate that the in-situ observations were located at ~750 hPa.

*We have added this to the abstract. "Using aircraft observations from the Colorado Front Range, we demonstrate that TES can be sensitive to elevated PAN in the boundary layer (~750 hPa) even in the presence of clouds."*

Main text:

*Note: The line numbers that the reviewer is citing refer to the version of the manuscript that was submitted with track changes noted.*

L290 – "Of all the retrievals attempted in July 2006 to July 2009, the percent associated with smoke is 18%. We expect a higher fraction of overlap in the subset of data with DOF > 0.6. (XX%)" – Please add your quantitative findings here.

*As the sentence originally indicated, there is a higher fraction of overlap in this subset of data, and we have added the percentage to the sentence. It now reads:*

*"We expect a higher fraction of overlap in the subset of data with DOF > 0.6 (28%)."*

Minor comments:

L137 TES is capable of measuring elevated PAN (detection limit ~ 0.2 ppbv) in the free troposphere, with uncertainty of 30-50 %. Citation? Validation?

*This sentence refers to Payne et al. (2014), similar to the sentences above. We have added a reference to this sentence.*

*"On a single footprint basis, TES is capable of measuring elevated PAN (detection limit ~ 0.2 ppbv) in the free troposphere, with uncertainty of 30-50 % (Payne et al., 2014)."*

L170-189 – are these findings broadly consistent with what we already know? please include citations

*We have added the following sentences to the paper:*
*"The increase with latitude has also been observed by TES over the eastern Pacific Ocean (Zhu et al., 2017), and GEOS-Chem also produces a similar pattern in the mid-troposphere (Fischer et al., 2014). We are unaware of other work that has examined a longitudinal gradient in PAN over North America. PAN has recently measured by in situ instruments in the Colorado Front Range, and mixing ratios exceeding 1 ppbv do occur in this region (Zaragoza et al., 2017)."*

L204 - In the case study presented, the PAN enhancement is located at ~750 hPa. I presume the author's comment "below 800 hPa" refers to larger pressures, or lower altitudes.

*Yes, this is what we intend here. We have made a small edit to the sentence to make that clear. This now reads: "The red and blue lines show that TES has some sensitivity to PAN located at altitudes below 800 hPa, but the retrieval places the additional PAN higher up in the atmosphere."*

L318 – "We show the PAN distribution for in-smoke cases that also coincide with TES 510hPa CO > 120 ppbv and TES 510hPa CO > 150 ppbv. There are differences between these subsets of data and the not-in smoke cases." To make the above claim that PAN retrievals are enhanced in smoke, and not just enhanced where CO is enhanced, you should control for CO retrievals in both in-smoke and not-in-smoke categories.

*We removed the sentence that the reviewer is referring to in this comment. We also agree*

*that this is a great idea to pursue, so we tried! However, there are very few (17) TES CO retrievals with 510hPa CO > 150 ppbv that are not located within smoke polygons. So doing a quantitative comparison of this subset, which we believe is the most conservative indication of the presence of smoke where TES is sensitive to PAN, is not possible because of the low number. TES CO retrievals with 510hPa CO > 150 ppbv are almost always associated with smoke. Thus we have done a similar CO sub-setting to the "Not In Smoke" data for Figure 5a. We have also added the N to the figure because the caption was confusing as to which numbers applied to which subset. The updated text is included here.*

[revised manuscript text omitted]

Emily Fischer 3/1/2018 4:51 PM

Emily Fischer 3/1/2018 3:24 PM

Emily Fischer 3/1/2018 4:57 PM

Emily Fischer 3/1/2018 4:57 PM

Emily Fischer 3/1/2018 4:57 PM

[Figure]

**Figure 6:** Successful TES PAN retrievals overlapping NOAA HMS smoke polygons for July 2006 to July 2009 colored by the day of the month. Filled circles denote the set of retrievals that also coincide with 510 hPa CO greater than 150 ppbv. This set of point is used to calculated PAN enhancement ratios relative to CO in Figure 8.

[Figure]

**Figure 7:** Top panel: Case study of TES PAN retrievals overlapping HMS smoke polygons 22 – 23 July 2007. Orange triangles represent FIRMS MODIS Hotspots for 22 July (Product MCD14ML; https://firms.modaps.eosdis.nasa.gov/download/request.php). Blue triangles represent FIRMS MODIS Hotspots for 23 July. Red circles indicate TES PAN retrievals on 22 July, and purple circles represent TES PAN retrievals on 23 July. We have circled the two retrievals in this set with 510 hPa CO greater than 150 ppbv. The PAN enhancement ratios for these points are noted in Figure 7. The purple lines signify 5 day HYSPLIT backward trajectories initialized at each TES retrieval at 4 km. The purple '+' signifies 24 hours of transport time on the 4 km trajectories. The black dashed line shows the location of the CALIPSO swath shown in the lower panel. Lower panel: CALIPSO aerosol subtype observed on 23 July 2007. CALIPSO Science Team (2016), CALIPSO/CALIOP Level 2, Vertical Feature Mask Data, version 4.10, Hampton,

VA, USA: NASA Atmospheric Science Data Center (ASDC), Accessed by Emily V. Fischer at doi: 10.5067/CALIOP/CALIPSO/LID_L2_VFM-Standard-V4-10

[Figure]

**Figure 8:** Histogram of estimated PAN enhancement ratios based on tropospheric mean PAN and CO from July 2006 – 2009 North American TES PAN retrievals overlapping HMS smoke plume polygons. The solid black line represents enhancement ratios calculated using an assumed PAN background of 0.1 ppbv with an assumed CO background of 80 ppbv. The dotted black line represents enhancement ratios calculated using an assumed PAN background of 0.1 ppbv with an assumed CO background of 90 ppbv. These specific enhancement ratios were calculated using an assumed CO background of 80 ppbv, similar to the solid black line. The dashed line represents enhancement ratios calculated using a significantly higher assumed PAN background of 0.2 ppbv with an assumed CO background of 90 ppbv. In all cases, negative values are not shown. The purple dots are the enhancement ratios for the two circled retrievals on 23 July 2007 plotted in Figure 7 associated with transported smoke.

[Figure]

**Figure 9:** Histogram of estimated PAN enhancement ratios based on in situ measurements of fire plumes
described in Hecobian et al. (2011) from the ARCTAS campaign. Enhancement ratios were calculated
using the 25th percentile for each trace gas during the corresponding flight day. These ratios were calculated
using the 1-minute merged data.